# Competition between inside-out unfolding and pathogenic aggregation in an amyloid-forming β-propeller

Emily G. Saccuzzo[1,8], Mubark D. Mebrat[2,3,8], Hailee F. Scelsi[1], Minjoo Kim [2,3], Minh Thu Ma[1], Xinya Su[4], Shannon E. Hill[1], Elisa Rheaume[5], Renhao Li [6], Matthew P. Torres[4], James C. Gumbart[1,4,7], Wade D. Van Horn [2,3] ✉ & Raquel L. Lieberman [1] ✉

Studies of folded-to-misfolded transitions using model protein systems reveal a range of unfolding needed for exposure of amyloid-prone regions for subsequent fibrillization. Here, we probe the relationship between unfolding and aggregation for glaucoma-associated myocilin. Mutations within the olfactomedin domain of myocilin (OLF) cause a gain-of-function, namely cytotoxic intracellular aggregation, which hastens disease progression. Aggregation by wild-type OLF (OLF$^{WT}$) competes with its chemical unfolding, but only below the threshold where OLF loses tertiary structure. Representative moderate (OLF$^{D380A}$) and severe (OLF$^{I499F}$) disease variants aggregate differently, with rates comparable to OLF$^{WT}$ in initial stages of unfolding, and variants adopt distinct partially folded structures seen along the OLF$^{WT}$ urea-unfolding pathway. Whether initiated with mutation or chemical perturbation, unfolding propagates outward to the propeller surface. In sum, for this large protein prone to amyloid formation, the requirement for a conformational change to promote amyloid fibrillization leads to direct competition between unfolding and aggregation.

Approximately 20 folded and disease-associated proteins are known to form amyloid, often as a result of a non-synonymous coding mutation[1]. Studies of tractable model systems demonstrate that the resultant amino acid substitution in such proteins introduces some degree of unfolding, exposing amyloid-templating peptide (or amyloid-prone region, APR) sequences that are not readily accessible in the wild-type (WT) protein[2]. A recent addition to the list of pathogenic proteins that form amyloid is the myocilin olfactomedin (OLF) domain[3]. Many different single-point mutations in myocilin (*MYOC*, Uniprot accession Q99972), dispersed throughout the sequence and

structure, comprise the strongest genetic link to early onset open angle glaucoma[4]. Mutant myocilin hastens the causal risk factor of elevated intraocular pressure[5], which originates from dysregulation of the trabecular meshwork extracellular matrix[6] where myocilin is robustly expressed[7]. Early cellular studies revealed a pathogenic mechanism involving a toxic gain of function where cellular secretion was impaired and, instead, detergent-insoluble aggregates accumulated[8–13]. Loss of function was ruled out by the finding that individuals with heterozygous[14] and hemizygous[15] N-terminal truncation mutations do not develop glaucoma.

[1]School of Chemistry & Biochemistry, Georgia Institute of Technology, Atlanta, USA. [2]Biodesign Center for Personalized Diagnostics, Arizona State University, Tempe, USA. [3]School of Molecular Sciences, Arizona State University, Tempe, USA. [4]School of Biological Sciences, Georgia Institute of Technology, Atlanta, USA. [5]Interdisciplinary Graduate Program in Quantitative Biosciences, Georgia Institute of Technology, Atlanta, USA. [6]Aflac Cancer and Blood Disorders Center, Children's Healthcare of Atlanta and Department of Pediatrics, Emory University School of Medicine, Atlanta, USA. [7]School of Physics, Georgia Institute of Technology, Atlanta, USA. [8]These authors contributed equally: Emily G. Saccuzzo, Mubark D. Mebrat. ✉e-mail: wade.van.horn@asu.edu; Raquel.lieberman@chemistry.gatech.edu

Our understanding of pathogenic protein misfolding at the molecular level is limited in general[16], and our comprehension of OLF misfolding in particular is in its infancy. OLF is a 30 kDa, $Ca^{2+}$-bound 5-bladed β-propeller (Fig. 1)[17]. β-propellers do not have a canonical hydrophobic core. Instead, the radial arrangement of β-sheets composed of 4-antiparallel strands form a toroid that forms a molecular clasp from the extreme N- and C- terminal β-strands. β-propellers are stabilized by hydrophobic interactions between β-propeller blades that surround a central hydrophilic channel. Additional contacts with the blades occur with loops at the top and bottom face of a β-propeller[18]. In OLF, four of the five blades are further stabilized by ligand binding to a dimetal center in the central channel[17]. Bioinformatics analysis and experiments identified two APR sequences within OLF (Fig. 1): P1 ($G_{326}$AVVYSGSLYFQ) located in the two innermost strands of blade B and P3 ($V_{426}$ANAFIICGTLYTVSSY) comprising the two innermost strands of blade D. A third stretch, P2 ($G_{387}$LWVIYSTDEAKGAIVLSK) within blade C, was predicted to form amyloid but remained soluble when tested experimentally[19]. Pathogenic mutations can be defined biophysically as those with thermal stability ($T_m$)≤ 47 °C for their OLF domains[20], 5 °C lower than WT[21]. This decreased stability facilitates amyloid formation at 37 °C and in neutral pH buffers[3,19]. In the context of full-length myocilin in a cellular environment, disease variants form amyloid in the endoplasmic reticulum (ER)[20]. The ER Hsp90 paralog Grp94 coaggregates with mutant myocilin, obstructing ER-associated degradation[22]. Beyond mutant OLFs, WT OLF (OLF^WT) converts from a stable protein to an amyloid fibril under mildly destabilizing conditions in vitro, such as slightly elevated temperature or acidic pH[3,19], and full-length WT myocilin forms amyloid aggregates in cells upon calcium dysregulation[23].

In this study, we better comprehend the proclivity of OLF toward amyloid aggregation by probing the molecular details of its misfolding and aggregation using structural and biophysical tools. We use both low- and high-resolution techniques to probe the mechanism by which OLF^WT initially transitions from a folded to a partially unfolded, aggregation prone state, as well as the relationship between partially unfolded states adopted by OLF^WT and those that are adopted by disease-causing mutants. Fibrillization competes with chemical unfolding in OLF^WT, but only below the limit where the domain loses detectable tertiary structure. Two well-studied glaucoma variants representative of moderate (OLF^D380A) and severe (OLF^I499F) disease (Fig. 1)[19,24,25] aggregate at rates comparable to OLF^WT incubated with ~1 M urea, yet each adopts a unique partially folded, non-native structure in solution. In line with the requirement of internal APRs to be exposed for templated fibril formation to occur, protein unfolding is initiated with the loss of internal residues, which propagates

unfolding outward. OLF^WT exhibits unusual protein dynamics with limited undulations detectable in the picosecond to microsecond range and unprecedented, exceptionally slow dynamics on the timescale of weeks. Thus, the OLF^WT domain is not at equilibrium and more complex than well-studied model systems. We show that for large proteins prone to amyloid formation, unfolded intermediates that nucleate aggregation accumulate to an appreciable extent during unfolding, resulting in significant competition between the pathways of folding-unfolding and aggregation-fibrilization.

## Results

### Amyloid aggregation of OLF^WT competes with unfolding

To evaluate the relationship between unfolding and aggregation of OLF^WT, we monitored kinetics of amyloid formation in the presence of varying concentrations of urea (Fig. 2a, Supplementary Fig. 1, Supplementary Table 1). Consistent with prior work[3], no ThT-positive aggregates emerge when OLF^WT is incubated at 37 °C without urea, but increasing ThT-positive kinetics are seen from 0.25-3 M urea (Fig. 2a, Supplementary Fig. 1a, b, Supplementary Table 1). OLF^WT incubated with 6 M urea did not aggregate (Fig. 2a, Supplementary Fig. 1a, b, Supplementary Table 1). Representative samples selected and visualized with negative stain transmission electron microscopy confirm the presence of a donut-shaped oligomer and fibrillar species consistent with amyloid (Supplementary Fig. 1c) with similar morphologies to those obtained for OLF^WT previously[3]. Rate constants estimated from the slope of the linear portion of the growth curves (Supplementary Fig. 1a, Supplementary Table 1), plotted as a function of urea concentration, and fit to a Gaussian, demonstrate the maximal aggregation rate occurs with 2.5 ± 0.7 M urea (Fig. 2b). At room temperature, OLF^WT incubated in 3 M urea, where tertiary structure is no longer detected by near-UV circular dichroism (CD), can be refolded upon dilution to buffer lacking urea (Fig. 2c)[20]. By contrast, when OLF^WT incubated in 5 M urea is diluted, significant precipitation is visible. Still, some OLF^WT remains in solution, at a high enough concentration to be measured by near-UV CD, and this species does not have detectable tertiary structure (Fig. 2c). Thus, refolding OLF^WT from urea is possible from the midpoint of unfolding but not at later stages of unfolding.

Corroborating evidence for competition between amyloid aggregation and unfolding derives from monitoring unfolding of OLF^WT measured using intrinsic fluorescence. No change in unfolding midpoint is detected after freshly purified OLF^WT (Fig. 2d) or OLF^WT protein stored for ~2 months at 4 °C is incubated in urea for 1 h or 48 h (Supplementary Fig. 2). The midpoint of unfolding at 25 °C is 2.7 M urea. At 37 °C, the midpoint shifts to a lower concentration of 2.3 M, likely because at higher temperatures the fraction of unfolded protein is higher at any given urea concentration. The unfolding transition is also shallower at 37 °C than at 25 °C, indicating an accumulation of more unfolding intermediates than at 25 °C. Taken together, our data support the hypothesis that partially folded OLF^WT is primed for amyloid aggregation, which competes with OLF^WT unfolding in urea.

### Aggregation of OLF^WT in ~ 1 M urea, OLF^D380A, and OLF^I499F

Next, we compared aggregation rates for representative disease variants OLF^D380A and OLF^I499F [19] with OLF^WT in urea. OLF^D380A and OLF^I499F differ from OLF^WT in their calcium content. D380 was initially discovered as a $Ca^{2+}$ ligand[26] by showing that in the absence of EDTA in purification buffers, OLF^WT contained stoichiometric levels of calcium, and calcium binding was ablated in OLF^D380A. Analysis of $Ca^{2+}$ levels in OLF^I499F by inductively coupled plasma optical emission spectroscopy (ICP-OES) reveals ~20% occupancy compared to OLF^WT under buffer conditions used in this study and elsewhere[23,26] (Supplementary Table 2). Thus, despite still harboring all residues necessary for $Ca^{2+}$ chelation, the metal center in OLF^I499F is not fully occupied and thus predominantly non-native. The previously determined[19] aggregation rate constants for OLF^D380A and OLF^I499F are closest to that of OLF^WT in

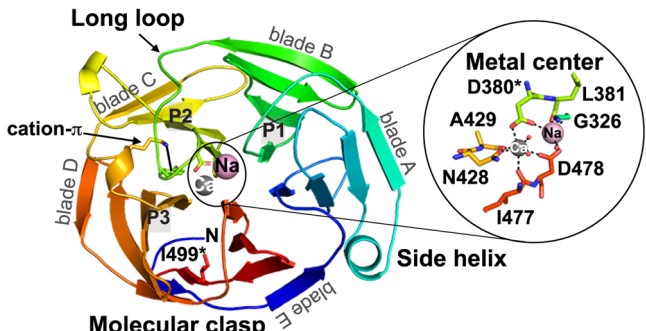

**Fig. 1 | The OLF^WT propeller structure and key regions of interest for amyloid formation.** Cartoon representation of OLF^WT (PDB code 4WXS, [https://doi.org/10.2210/pdb4WXS/pdb] from N (blue) to C (red terminus). $Ca^{2+}$, grey sphere. $Na^+$, pink sphere. P1, P2, and P3 are predicted APRs, of which isolated peptides comprising P1 and P3 sequences form amyloid. *, positions of mutations (D380, I499) described in the manuscript.

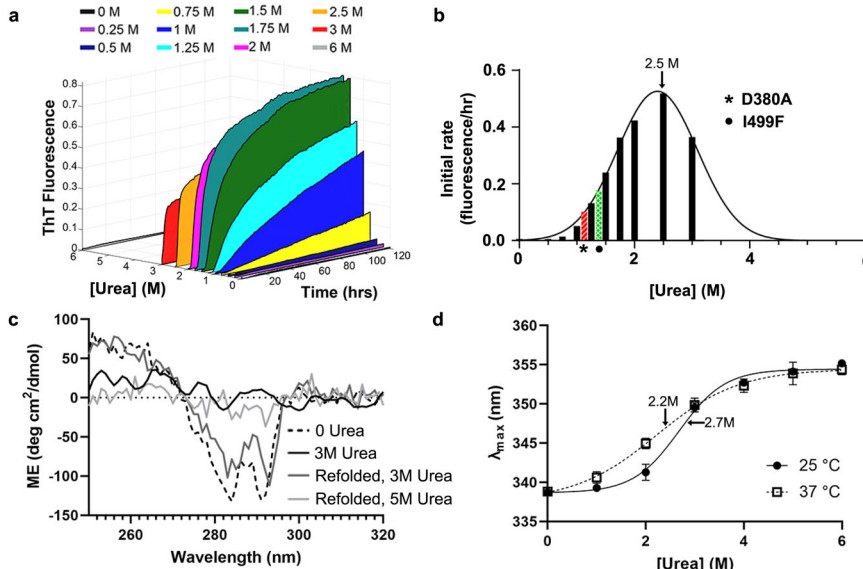

**Fig. 2 | OLF$^{WT}$ unfolding and aggregation as a function of urea concentration.** **a** Kinetics of aggregation tracked by ThT fluorescence increase as a function of urea concentration (M). See also Supplementary Fig. 1a, b, Supplementary Table 1b **Initial rates of aggregation from (A) increase until a midpoint of ~2.5 M is reached. Rates of representative disease mutants OLF$^{D380A}$ and OLF$^{I499F}$ fall on the curve for WT OLF unfolding in the range of 1-1.3 M urea. **c** Refolding experiments tracked by tertiary CD spectra demonstrate OLF$^{WT}$ can be refolded from 3 M urea but not 5 M urea. For both samples lacking tertiary structure, protein concentration in solution was sufficiently high for spectral acquisition. **d** Urea unfolding as a function of intrinsic fluorescence maximum emission wavelength ($\lambda_{max}$) at two different temperatures showing shallower slope at 37 °C. Each data point is the mean +/− SD ($n = 6$). See also Supplementary Fig. 2. Source data are provided as a Source Data file.

1.1 M and 1.3 M urea, respectively, as calculated from Fig. 2b. OLF$^{D380A}$ and OLF$^{WT}$ in the presence of ~1.1 M urea both exhibit a largely linear increase in ThT fluorescence over the course of the experiment (Fig. 2b, Supplementary Fig. 1d), whereas aggregation of OLF$^{I499F}$ is faster, most similar to the hyperbolic curve seen for OLF$^{WT}$ in the range of 1.33 M urea (Fig. 2b, Supplementary Fig. 1). Taken together, OLF$^{D380A}$ and OLF$^{I499F}$ exhibit rates of aggregation are mimicked by partially folded states of OLF$^{WT}$ accessed by urea.

## Structural fingerprints of OLF$^{D380A}$ and OLF$^{I499F}$ by HDX-MS

Hydrogen-deuterium exchange mass spectrometry (HDX-MS) experiments conducted on OLF$^{WT}$, OLF$^{D380A}$, and OLF$^{I499F}$ show some regions of similar and some unique deuterium uptake profiles (Supplementary Figs. 3–5, Supplementary Table 3). OLF$^{D380A}$ and OLF$^{I499F}$ show higher overall deuterium uptake after 10,000 s than the maximal uptake for OLF$^{WT}$ suggesting both variants are more dynamic than OLF$^{WT}$. The peptide regions experiencing enhanced uptake and rates of uptake in OLF$^{D380A}$ and OLF$^{I499F}$ differ from each other, however. Perhaps due to the central location of D380, the increased uptake OLF$^{D380A}$ is distributed across the sequence and structure, with several discontinuous peptide stretches reaching maximal net H/D uptake in the experiment. By contrast, OLF$^{I499F}$ the region that experiences maximal deuterium uptake is concentrated near the mutation at the C-terminus, in Blade E. Across all three proteins, Blade C experiences the least net deuterium exchange. Taken together, OLF$^{D380A}$ and OLF$^{I499F}$ appear to represent distinct partially folded states of OLF.

## OLF$^{WT}$ NMR backbone resonance assignment

To investigate the atomic-level differences among OLF$^{WT}$, disease variants OLF$^{D380A}$ and OLF$^{I499F}$, and partially unfolded OLF$^{WT}$ in urea, we employed solution NMR. The 2D $^1$H-$^{15}$N TROSY-HSQC spectrum of OLF$^{WT}$ is consistent with a well-structured protein (Fig. 3a). The HSQC data identify 285 dispersed resonances in close agreement with the number of residues in our WT OLF construct (277 amino acids), of which 13 are proline residues and thus not observable by $^1$H-$^{15}$N HSQC. To assign the OLF$^{WT}$ backbone resonances, we acquired $^1$H, $^{15}$N, C$_\alpha$, C$_\beta$, and C$_O$ chemical shift information from 3D TROSY-based NMR

experiments of uniformly double-labeled ($^{13}$C, $^{15}$N) OLF$^{WT}$. Analysis of these data resulted in 55% assignment of the OLF backbone (Fig. 3b, c, Supplementary Fig. 6, BMRB ID: 51750 [https://doi.org/10.13018/BMR51750]). The assigned residues are distributed throughout the protein sequence and include some terminal residues absent in OLF$^{WT}$ crystal structures, specifically E230-S231, L236-T243 and K503-M504 (Fig. 3b, c, Supplementary Fig. 6).

## Analysis of OLF$^{WT}$ urea unfolding by NMR

Compared to the well-dispersed proton chemical shifts (6.3-11.0 ppm; Fig. 3a) in the $^1$H-$^{15}$N TROSY-HSQC spectrum of OLF$^{WT}$, increasing urea concentration results in $^1$H dimension chemical shift dispersion collapse (Fig. 4a, Supplementary Fig. 7). At 1 M and 2 M urea concentrations, the spectra are still generally dispersed (Supplementary Fig. 7) and many resonances overlap with the spectrum of OLF$^{WT}$, but some resonances experience significant chemical shift perturbation (CSP). Mapping of backbone assignment from the OLF$^{WT}$ spectrum to 1 and 2 M urea was evaluated quantitatively to minimize bias and over-interpretation (see Methods). Omitting resonances that disappeared or had large chemical shift changes resulted in 11 fewer assigned residues in the 1 M urea spectrum and 28 fewer assigned residues in the 2 M urea spectrum. Spectral comparison of OLF$^{WT}$ with 1 M and 2 M urea indicates that native or native-like structures dominate the conformational ensemble at low urea concentration as evidenced by broad $^1$H resonance dispersion (Supplementary Fig. 7) and is consistent with stable β-sheet character. High concentrations of urea (3-6 M) destabilize the folded state, and unfolded states dominate the conformational landscape, as evidenced by the collapse of $^1$H chemical shift dispersion. Principal component analysis (PCA) of the NMR-detected urea titration was used to quantify urea-dependent OLF chemical unfolding (Supplementary Table 4). The first principal component (PC1) was fit to a two-state unfolding model yielding a midpoint of OLF urea of $2.7 \pm 0.3$ M (Fig. 4b), in line with the midpoints of urea unfolding obtained using intrinsic fluorescence measurements (Fig. 2d) as well as the for maximum of ThT aggregation kinetics (Fig. 2b).

NMR assignments that can be confidently mapped from OLF$^{WT}$ to either 1 or 2 M urea, where the folded structure dominates the OLF

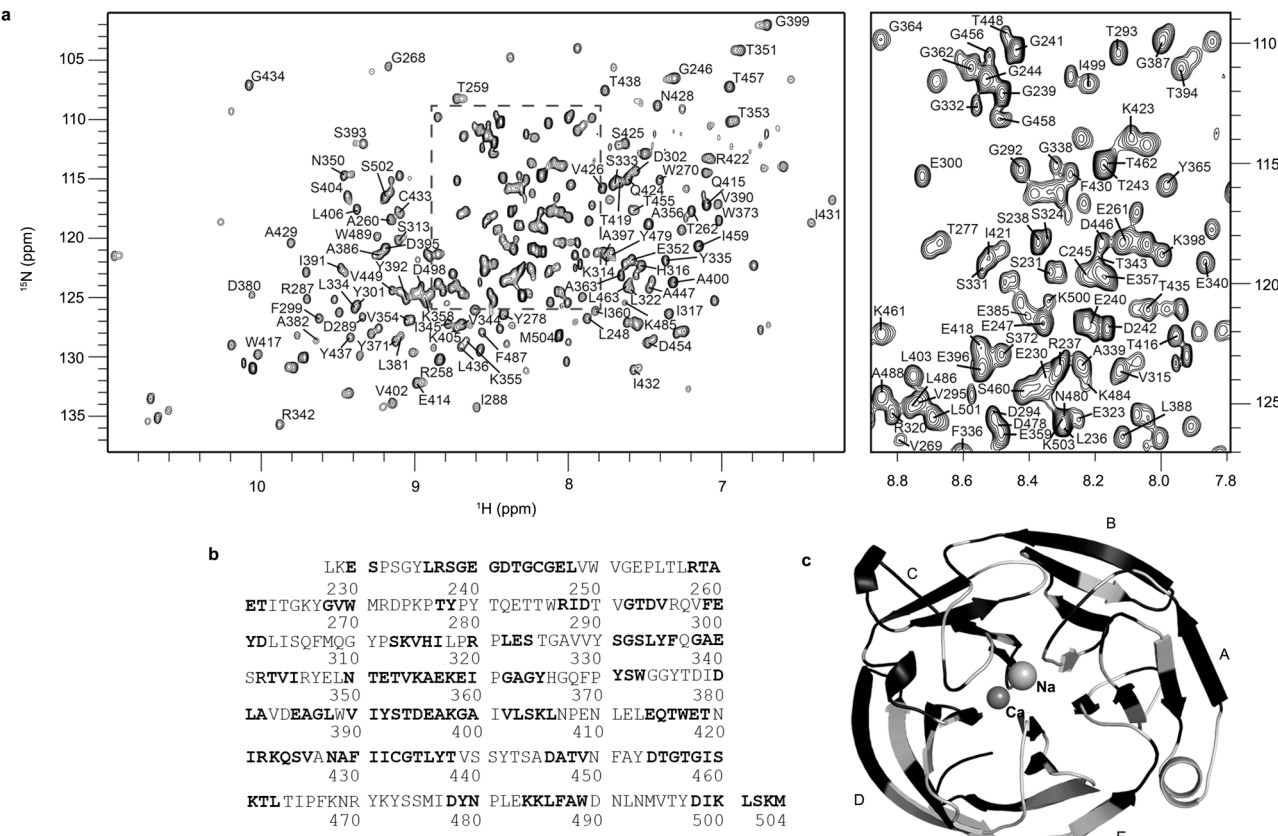

**Fig. 3 | Backbone solution NMR assignments of OLF^WT. a** ¹H-¹⁵N TROSY-HSQC spectrum labeled with assigned resonances. Right, close up of the dashed region from the left spectrum. **b** WT OLF sequence with assigned backbone resonances highlighted by bold. **c** Data from panel **b** mapped onto OLF^WT structure (PDB 4WXQ, [https://doi.org/10.2210/pdb4WXQ/pdb]) where black highlights assigned backbone resonances. See also Supplementary Fig. 6.

conformational ensemble, were used to evaluate OLF regions resistant to chemical unfolding. Mapping the chemical shift perturbations (Δδ ppm) between OLF^WT and 1 M urea (Fig. 4c, top) onto the structure identifies that residues with significant resonance changes, large shifts or disappearance appear to be primarily confined to the center of the OLF structure (Supplementary Fig. 8). Comparatively, the 2 M urea spectrum exacerbates the trend, with further significant resonance changes focused on the OLF propeller hub and radiating to the exterior of the blades (Fig. 4c, bottom, Supplementary Fig. 8).

### Analysis of the moderate disease mutant OLF^D380A by NMR

The ¹H-¹⁵N TROSY-HSQC spectrum of OLF^D380A has well-dispersed resonances, including many that overlap with the OLF^WT spectrum (Fig. 5a). These spectral similarities indicate that OLF^D380A primarily retains the β-propeller structure native to WT. Nevertheless, there are clear differences, including reduced resonance numbers, more heterogeneous resonance lineshapes, and a decrease in proton dimension dispersion in the OLF^D380A spectrum. Changes in the OLF^D380A spectrum are consistent with enhanced OLF dynamics compared to OLF^WT, which are slower on the NMR timescale (>ms) and a more diverse conformational ensemble, in line with results from HDX-MS, in which uptake for OLF^D380A is somewhat higher than for OLF^WT across the sequence (Supplementary Figs. 3–5).

Assignable residues in the OLF^D380A spectrum are generally distributed across the five propeller blades but CSP mapped onto the OLF^WT crystal structure reveals specific structural perturbations (Fig. 5b). In total, 73 resonance assignments were confidently transferred from OLF^WT to the OLF^D380A spectrum, which represents about 60% of the discreet OLF^D380A resonances (see Methods). First, the spectrum confirms previous knowledge that calcium binding is

ablated in D380A[26]. Besides D380, which was mutated to Ala, Ca²⁺ chelating residues N428, D478, and A429, are missing in the OLF^D380A spectrum. The final calcium ligand, I477, was not assigned and thus cannot be evaluated. Second-shell Ca²⁺ coordinating residues are also perturbed. These include residues in the inner two strands of blades C and D adjacent to the remaining ligands D478 and A429, such as Y437. Y371, on the long loop that forms an H-bond with D380 in OLF^WT, is also perturbed. Second, eight residues in the previously identified[19] APR stretch P3 (N428, A429, F430, I431, C433, L436, Y437, T438), experience CSP. Interestingly, P1 residues remain largely WT-like, indicating they retain a chemical environment that is similar to OLF^WT. This observation may explain why D380A forms P3-like fibrils[19]: if P3 residues are mobile and disordered and not sequestered as in WT, they are available to template fibril formation.

### Evaluation of the severe disease mutant OLF^I499F by NMR

In contrast to the WT and OLF^D380A spectra, the OLF^I499F ¹H-¹⁵N HSQC spectrum has few distinct resonances concomitant with largely collapsed proton resonance dispersion (Fig. 5c). These spectral features are consistent with the observation that the severe variant OLF^I499F adopts a modestly folded and dynamic structural ensemble that deviates significantly from the structure of OLF^WT. Differences in deuterium uptake in HDX-MS between OLF^WT and OLF^I499F did not capture the extent of the structural changes apparent by NMR; even though there are isolated regions of higher or lower apparent uptake in OLF^I499F than OLF^WT, many peptide stretches in the structurally resolved portion of the OLF domain did not experience net exchange at a rate distinguishable from OLF^WT (Supplementary Figs. 3–5).

Few of the assigned OLF^WT resonances overlap and thus can be transferred to OLF^I499F. Specifically, of the 106 discreet NMR

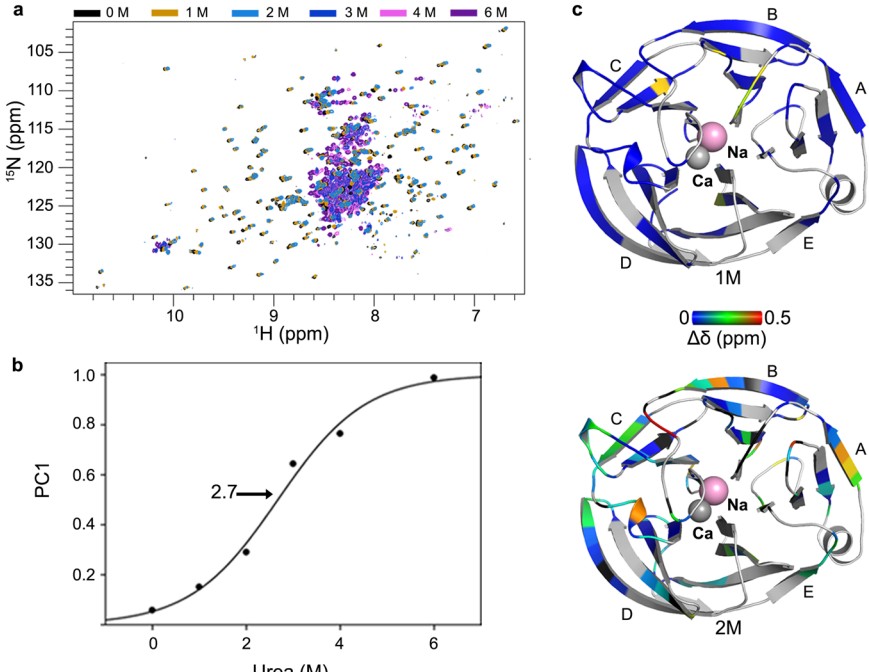

**Fig. 4 | OLF^{WT} chemical unfolding tracked by NMR. a** ^1H-^15N TROSY-HSQC data of OLF^{WT} as a function of urea concentration shows a transition from well-folded to denatured. **b** Principal component analysis of the data in panel **a** quantifies the midpoint of urea unfolding, 2.7 ± 0.3 M. **c** CSP of retained assignable resonances mapped onto OLF^{WT} structure from 1 M (top) and 2 M (bottom) spectra from panel (**a**) (mustard and blue spectra, respectively). These data identify regions of OLF that retain WT-like structure as they initially transition towards unfolding. At urea concentrations >2 M, the unfolded state is dominant and limits the ability to confidently map assignments and track CSP. Unassigned residues are in grey. See also Supplementary Figs. 7 and 8. Source data are provided as a Source Data file.

resonances in the OLF^{I499F} spectrum, 51 resonance assignments could be confidently transferred. Nevertheless, a local effect of OLF^{I499F} substitution on blade E can be deduced. Resonances for D498 and its salt bridge partner K485 on an adjacent internal strand were not transferrable. This effect appears to propagate further, as neighboring F487 and its π-π stacking partner W270 on blade A were likewise not transferrable. Structural changes due to the introduction of F499 appear to propagate to the other resonances experiencing detectable shifts within blade A and possibly further across the closed-circular OLF propeller (Fig. 5d, Supplementary Fig. 8). Consistent with metal analysis showing that as-purified OLF^{I499F} retains 0.2 Ca^{2+} per mol OLF^{I499F} (Supplementary Table 2), residues participating in the metal center appear perturbed. Analogous to OLF^{D380A}, calcium ligand D380 and second shell residue Y371 are not transferrable. In addition, the signals for D478 and A429 are not transferrable (Supplementary Fig. 8). The residual Ca^{2+} detected by metal analysis (Supplementary Table 2) may be bound to N428, the only assigned residue whose resonance was minimally perturbed. In contrast to OLF^{D380A}, APRs associated with amyloid formation were affected similarly, with three residues disappearing each within P1 (L334, Y335, F336) and P3 (A429, C433, Y437).

Despite these perturbations, 21 residues, distributed through the OLF sequence and structure, experienced a negligible change in ^1H-^15N chemical shift ( < 0.1 ppm), indicating some structural features of OLF^{WT} remain intact (Fig. 5d, Supplementary Fig. 8). For example, the triad S393, A400 and T419 on adjacent strands in blade C are present, and they form a H-bonding network in OLF^{WT}. Similarly, in blade D, interacting residues D454 and I459 are WT-like. Overall, the OLF^{I499F} residual structure appears largely non-native but some residual structural features remain native-like.

**NMR comparison of OLF^{D380A}, OLF^{I499F} and urea-treated OLF^{WT}**
Comparisons between the spectra of OLF^{D380A} and OLF^{I499F} disease mutants with partially chemically unfolded OLF^{WT} at 1 and 2 M urea show spectral similarities (Fig. 5e, Supplementary Figs. 8, 9). In particular, global fitting of Euclidean distances between the OLF^{D380A} chemical shift positions to those of the OLF^{WT} and urea NMR spectra indicate that OLF^{D380A} quantitatively most resembles the 1 M urea spectrum, in agreement with the aggregation kinetic rate constants (Fig. 2b), which closely resemble that of OLF^{WT} in 1.1 M urea. Even though the spectrum of OLF^{I499F} cannot be compared quantitatively to OLF^{WT} in urea, by analogy, the spectrum is consistent with increased conformational heterogeneity and the kinetic rate constant was more similar to 1.33 M urea (Fig. 2b), where OLF^{WT} is more unfolded. Taken together, NMR data are also consistent with the hypothesis that OLF fibrillization proceeds via access to multiple, and dissimilar, partially unfolded states.

**Dynamics of OLF^{WT} from ps-month time scale**
Next, we probed motions in OLF^{WT} that could be relevant to aggregation-prone partial unfolding. First, to probe picosecond time scale motions, we solved a 1.27Å crystal structure of OLF^{WT} (Fig. 6a, Supplementary Table 5). The new structure shares overall features with our previously reported ~2.1Å resolution structure (r.m.s.d < 0.3Å) solved from a pseudo-merohedrally twinned crystal[17], but the newly refined structure has a low B-factor range (10-30 for Cα, average 15) and limited anisotropic translation/libration/screw (TLS) motions (Fig. 6a). Furthermore, structural refinement using a multiple conformer algorithm is largely consistent with TLS and B-factor analysis, with fewer side chain conformations modeled in the center of the protein than on the surface (Fig. 6a). Overall, the main motions in the near-atomic resolution crystal structure are seen in alternative side chain conformations and the external strands of blades B, C, and D.

Second, to detect nano-to-microsecond motions, we carried out atomistic molecular dynamics (MD) simulations on OLF^{WT} (Fig. 6b, Supplementary Fig. 10). The simulations reveal relatively limited motions over a 10-μs trajectory, similar to previous results obtained carried out at higher simulation temperature albeit for a shorter time[27]. The largest root mean squared fluctuations (RMSF) values observed in

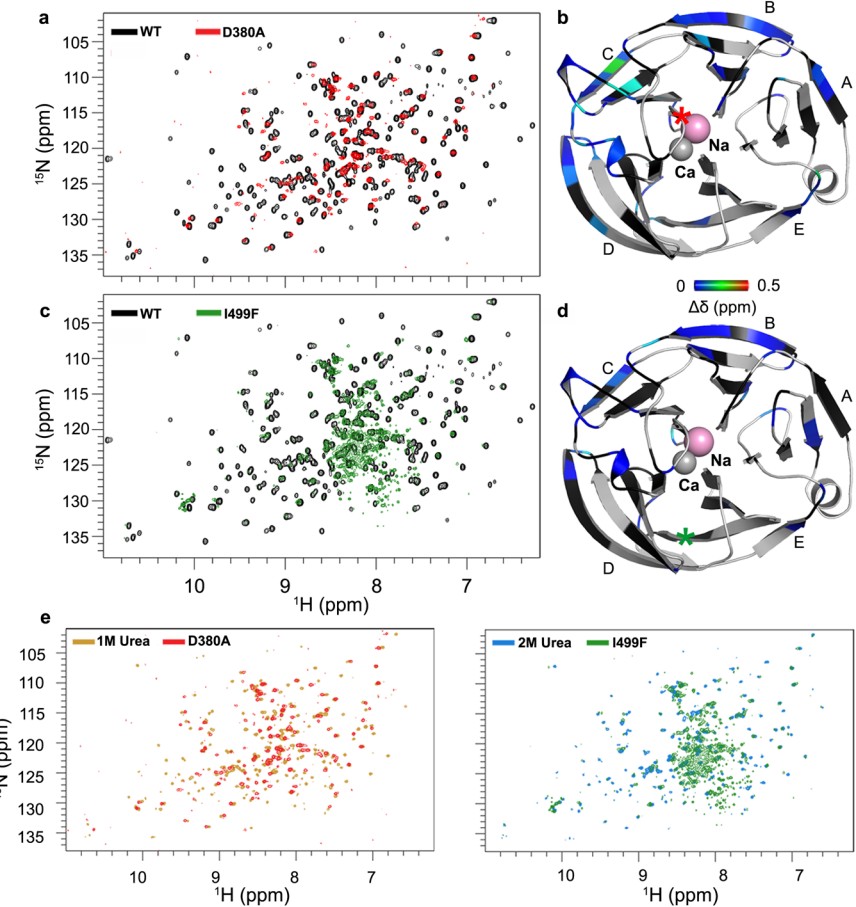

**Fig. 5 | Representative pathogenic mutants OLF^D380A and OLF^I499F are partially unfolded in solution. a** ^1H-^15N TROSY-HSQC of OLF^D380A (red) shows that, while this mutant retains significant folded β-sheet character from the ^1H resonance dispersion, OLF^D380A is more conformationally heterogeneous than OLF^WT (black). **b** Mapping regions in OLF^D380A that retain WT-like structure and assessing the CSP is colored from blue to red for increasing resonance position changes. Regions of the structure that are colored gray are not assigned and those colored black are missing relative to the WT spectrum, indicating obvious differences between mutant and folded WT

structures and/or dynamics. Location of residue D380 is indicated with a red star. **c** ^1H-^15N TROSY-HSQC spectrum of OLF^I499F (green) identifies that this mutant represents a predominantly unfolded conformation. **d** Analysis of structural retention and chemical shift perturbation in OLF^I499F identifies regions most altered by the disease mutant. Coloring is the same as in (**b**). Location of residue I499 is indicated with a green star. **e** Comparison of OLF^D380A and OLF^I499F with 1 M and 2 M urea OLF spectra show that mutants share spectral features with the chemically partially unfolded state. See also Supplementary Figs. 7, 8. Source data are provided as a Source Data file.

MD simulations are in loops of blade A (Fig. 6b, Supplementary Fig. 10). One additional loop in Blade D and the long loop connecting blades C and D appear somewhat mobile in the simulation (Fig. 6b, Supplementary Fig. 10). Some of these motions have been captured previously in OLF^WT crystal structures solved from crystals belonging to different lattices[17] as well as stable variants containing a mutation to prevent calcium binding at ligand D478 alone or in combination with mutation of other metal binding ligands (Supplementary Fig. 11)[17]. Conversely, blades B, C, and D remain largely unchanged throughout the simulation, in agreement with experiments presented here comparing elements that remain intact in OLF^WT partially unfolded with urea (Fig. 4c, Supplementary Fig. 8). Compared to MD simulations of OLF^WT, and in general agreement with HDX-MS experiments (Supplementary Fig. 3), OLF^D380A and OLF^I499F exhibit both a greater degree of fluctuations as well as unique fluctuations (Supplementary Fig. 10). Increased fluctuations are most notable in blades A and E in OLF^D380A. Simulations of OLF^I499F reveal fluctuations similar to, or even slightly less than, OLF^WT (Supplementary Fig. 10). Despite this trend, we note that while I499 stably participates in a hydrophobic cluster including L248, V251, I432, and L486 in the OLF^WT simulations, F499 breaks out of it. Overall, less fluctuation in the simulation is consistent with less net H/D exchange observed for OLF^I499F compared to OLF^WT. The main increase in fluctuations in OLF^I499F compared to OLF^WT is a loop within

Blade A, located on the bottom face; this feature is also seen in the OLF^D380A simulation, and in HDX-MS for both mutants compared to OLF^WT (Supplementary Fig. 3). Across simulations, the metal center is often perturbed as well (Supplementary Fig. 10). For example, we observe a greater average RMSF in residues comprising P3 either coordinating or near the ions (specifically N428, A429, and F430) in OLF^D380A compared to OLF^WT, in agreement with CSP observed in this region (Fig. 5b). Our inability to observe further structural changes may be due to limited (10-μs) sampling of the conformational landscape; it may also indicate that mutations alter the folding pathway, which is not explored in simulations starting from the OLF^WT folded structure. In summary, on the microsecond time scale simulations suggest fluctuations within OLF are limited largely to loops and the metal center, with distinct perturbations for each mutant.

Unexpectedly, while attempting NMR-based OLF dynamics investigations, we discovered an unprecedented months-long change in ^1H-^15N-HSQC spectra of OLF^WT. Spectra acquired over the course of several months reveals that OLF^WT converts from an initial folded conformational ensemble to a final one that is also highly folded and retains the general OLF^WT β-propeller structure (Supplementary Fig. 12a). PCA of the subtle spectral changes indicates that the kinetic process has a half-life of 23 ± 3 days (Fig. 6d, Supplementary Table 6). Six assigned resonances (S231, G239, G241, G246, V315, G456) undergo

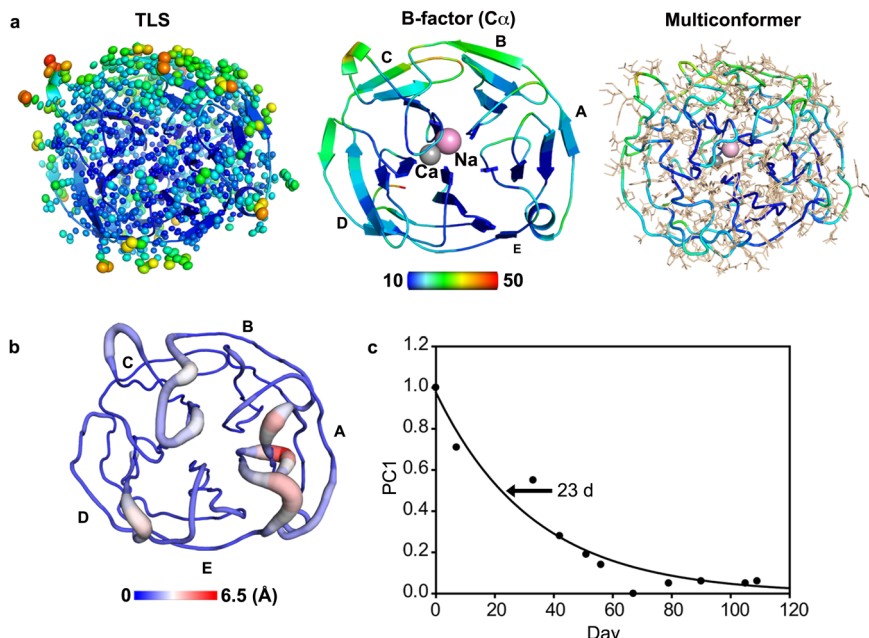

**Fig. 6 | OLF^WT exhibits unusual dynamics across multiple timescales. a** Analysis of picosecond time scale motions from TLS refinement (left, ellipsoids), final refined B-factors (middle, Cα range 10–50 blue-to-red rainbow), and multi-conformer side chain fitting and refinement (right, Cα ribbon colored by B-factor with beige side chains) of 1.27 Å resolution crystal structure of OLF^WT (PDB code 8FRR, [https://doi.org/10.2210/pdb8FRR/pdb]). See also Supplementary Table 4. **b** Nano-to-microsecond timescale dynamics from MD simulations with RMSF mapped onto the structure (PDB code 4WXS, [https://doi.org/10.2210/pdb4WXS/pdb], range 0–5 blue-to-red rainbow). See also Supplementary Fig. 10. **c** month timescale dynamics from NMR. See also Supplementary Figs. 12, 13. Source data are provided as a Source Data file.

the most significant (>3σ) CSP. Four are Gly, perhaps due to their inherent conformational flexibility. Three residues (S231, G239, and G241) are located at the N-terminus that links the OLF domain to a leucine zipper domain[28] in full-length myocilin and are not observed in any OLF^WT crystal structure, whereas the remaining three residues are distributed in blades A, D, and E.

The finding that OLF^WT remains globally unchanged despite the subtle change in the NMR spectrum is supported by a number of experiments. There are no changes in the unfolding properties of OLF^WT incubated at 4 °C for several months (Supplementary Fig. 2), nor are there changes in the crystal structure described above (Fig. 6a), which was solved from 10 month old crystals (see Methods). Denaturing SDS-PAGE analysis of the NMR sample after more than 3 months shows no evidence of proteolysis (Supplementary Fig. 6a versus Supplementary Fig. 12b). Further, comparing the post-translational modifications (PTMs), namely, deamidation and methionine oxidation, in the sample used for NMR that had been stored at 4 °C for over a year and freshly purified OLF^WT (Supplementary Fig. 13) reveals just one PTM detected exclusively in the old sample, deamidation at N480, but it was detected at very low abundance (2.22% mean modified %PSM). Eleven PTMs were detected in both samples. The extent of methionine oxidation or deamidation across these common sites in the two samples differ by <10% in most cases except for N350, N450, and M494 where the difference is less than 17%. In sum, the exceptionally long half-life for the observed structural change cannot be reconciled by cleavage, degradation, or PTMs and suggests that OLF^WT possesses two very similar structural states that are separated by a high energetic barrier. Consequently, the OLF β-propeller exists primarily in a non-equilibrium state on the physiological timescale.

## Unfolding from the inside-out primes OLF for amyloid aggregation

Taken together, our data are consistent with radial unfolding of OLF, with the exterior of the blades being most resistant to unfolding. First,

comparison of OLF^WT in 1 and 2 M urea indicates that an initial trigger for unfolding is the destabilization of central residues (Fig. 7a, Supplementary Fig. 8). Of the assigned resonances in the 1 M urea spectrum that are lost within the propeller blades, none is in outer strands and two are proximal to the calcium center metal ligand D478 and A382, which is adjacent to ligand L381 (Fig. 1). Comparison of OLF^WT spectra in 1 M versus in 2 M urea shows continued polypeptide unfolding, propagating from internal resonances initially lost in 1 M urea (Fig. 7a). For example, residues adjacent to D478 in blade E and adjacent blade A experience increased CSP at 2 M compared to 1 M (Fig. 7a, Supplementary Fig. 8). Interestingly, other calcium-coordinating ligands like D380 and N428 experience minimal perturbations in both the 1 and 2 M spectra. The most likely interpretation of this is that Ca^{2+} binding are retained to some extent during urea-induced unfolding, a notion that is supported by MD simulations of OLF^I499F and OLF^D380A in which metal ions remain at least partially bound (Supplementary Fig. 10) as well as by crystal structures of certain metal ligand stable variants tested (Supplementary Fig. 11) in which a metal binds, but using an alternative ligand arrangement. Second, CSP trends among OLF^WT, OLF^D380A, OLF^I499F, and OLF^WT incubated with 1 or 2 M urea reveals that the structural features retained across all four proteins are primarily surface-exposed, particularly within blades C and D (Fig. 7b, Supplementary Fig. 8). These regions coincide with areas of OLF^WT found to experience some undulations in the picosecond time scale, but limited exchange at the microsecond time scale (Fig. 6a–c, Supplementary Fig. 10). Retention of any tertiary structure is particularly notable for OLF^I499F because it is the least stable disease variant (T_m = 45.0 ± 1.1 °C) that we can isolate and purify in quantities required for detailed biophysical studies[19], and its chemical dispersion is limited in NMR spectra (Fig. 5c). In OLF^D380A, where calcium binding is ablated by mutation, all but one of the assigned resonances within the innermost strands of each blade are lost. In contrast, surface residues are perturbed to a lesser extent (Supplementary Fig. 8). In summary, our data suggest a model that explains how OLF misfolding arises, from inside out by unfolding β-propeller

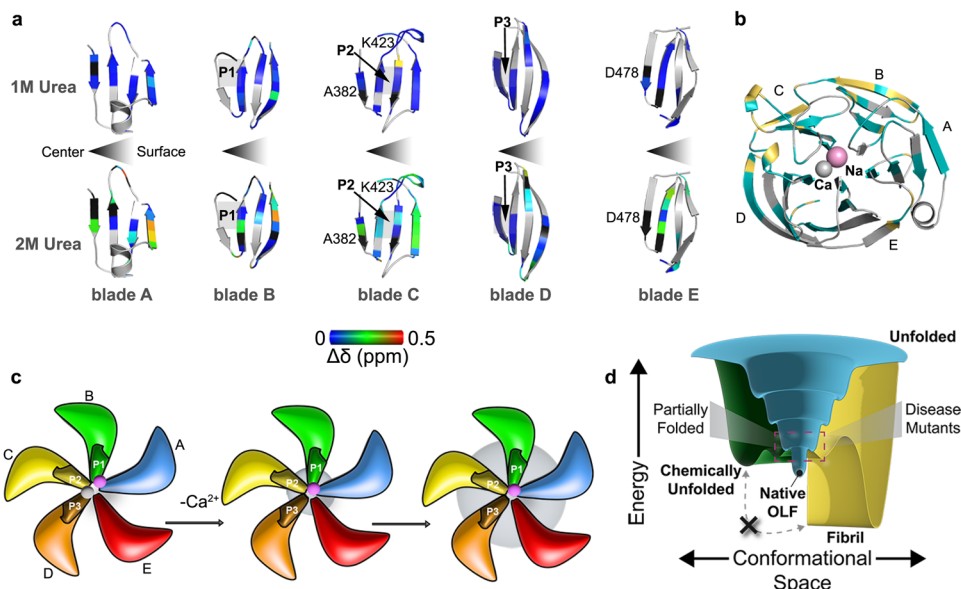

**Fig. 7 | OLF unfolding from the inside-out. a** Blade-by-blade comparison of CSP for 1 M and 2 M urea shows missing residues (black) are localized to the OLF center. Blue-to-red rainbow indicates 0-0.5 Δδ (ppm). See also Supplementary Fig. 8. **b** Yellow, highlights residues that were present, aka structured and with minimal CSP from OLF$^{WT}$ to OLF in 1 M urea, OLF in 2 M urea, OLF$^{D380A}$, and OLF$^{I499F}$. The yellow-highlighted resonances are primarily localized to blades B and C. Teal highlighted resonances were assigned in OLF$^{WT}$ but not transferred across all samples. See Methods for resonance transfer criteria. **c** Cartoon model of inside-out unfolding, which starts with loss of Ca$^{2+}$ and propagates outward. Each blade is differentiated by color (A, blue; B, green; C, yellow; D, orange; E red). Candidate APRs P1, P2 and P3 indicated (see also Fig. 1). Increasing radius of grey circle indicates extent of unfolding. **d** Energy surface diagram model of competition between fibril and unfolding pathways for OLF. Folding funnel, blue; chemical unfolding pathway, green; fibrillization, yellow.

(Fig. 7c), which differentially exposes internal APRs on the pathway to fibrillization (Supplementary Fig. 8).

## Discussion

The molecular mechanism of fibrillization is protein-dependent, but is generally thought to involve the association of APRs within a partially unfolded protein to form soluble oligomers until a nucleating threshold is reached, after which a mature fibril grows and becomes insoluble[29]. Studies of numerous model systems have provided key insights into how a partially folded, fibrillization-prone state of a folded protein can be accessed. For lysozyme, amyloidogenic partial unfolding can occur with chemical denaturants[30,31], heat[32,33], or mutation[29]. Similarly, experimental conditions, including low pH and mutation, dictate access to partially folded β2-microglobulin (B2M), and the specificity of a given treatment can affect the morphology of the amyloid structure obtained[34]. Transthyretin and the immunoglobulin light chain variable domain SMA first undergo amyloid fibril formation after dissociating from a tetramer[35] or dimer[36] to monomer, respectively, which in turn initiates other misfolding events. In SOD1, it is the initial loss of bound metal ions that causes dissociation of the native dimer to monomer, followed by additional conformational changes that enables access to an amyloidogenic partially folded SOD1[37]. The OLF domain of myocilin, a newcomer to the list of proteins capable of forming amyloid and associated with human disease, is an attractive case to explore themes that have emerged from studies of smaller model amyloidogenic protein systems. Our study reveals numerous ways in which OLF aggregation is more nuanced than better-studied model systems.

Given that OLF is 30 kDa, the finding that its unfolding differs from the ideal two-state model often ascribed to smaller proteins[38] is not unexpected. The shallow slope of OLF$^{WT}$ unfolding in urea suggests the presence of intermediates in the conversion of OLF$^{WT}$ from a stable folded protein to a chemically unfolded one at 37 °C. OLF$^{WT}$ is resistant to thermal denaturation in the absence of urea, indicating a relatively high degree of kinetic stability[39] in the absence of perturbing conditions. The finding that OLF unfolding is non-two-state is in line with protein design studies of β-propeller folding that document folding intermediates[40], perhaps due to their evolution via the duplication of individual blades[41].

Our data converge on a model in which OLF unfolding begins from internal residues and propagates outward. The molecular clasp likely experiences perturbations given that D478 within blade E is one of the early residues perturbed with 1 M urea. We also infer that the clasp may be somewhat more resistant to complete unfolding due to a disulfide bond between C433 on blade D and C245 on blade E but note we do not have complete resonance assignments in this region. The side helix that sits between blades E and A, which likewise remains unassigned, may also be somewhat unwound based on our structures of metal ligand variants (Supplementary Fig. 11) but this can happen without introducing thermal destabilization characteristic of aggregation-prone disease variants[27,42]. Overall, we deduce that perturbations yielding destabilized, aggregation-prone OLF are distinct from what is observed for stable variants that do not aggregate.

The common feature of inside-out unfolding, in spite of unique partially folded states for OLF$^{D380A}$ and OLF$^{I499F}$, rationalizes that exposure of internal APRs P1 and/or P3 trigger templated OLF fibrillization. With this knowledge, we can focus future efforts on the details of the connection between chemical unfolding and metal binding in OLF$^{WT}$, as well as between misfolding and metal ion binding properties of additional pathogenic mutants, to further our understanding of OLF unfolding. Finally, these observations raise the possibility that stabilizing neighboring residues to avoid unfolding could prevent aggregation. For example, stabilization with a small molecule or antibody[43], might be an effective therapeutic approach to abort pathogenic aggregation associated with mutant myocilin.

Conversely, our data show certain structural features, particularly in blades B and C, are retained across all partially folded samples characterized in this study. A notable surface-exposed cluster of five residues, K423 and four neighboring surface residues within P2 (Fig. 1), remain present. This region is also largely unchanged across OLF$^{WT}$,

OLF$^{D380A}$, and OLF$^{I499F}$ in the HDX-MS experiments (Supplementary Fig. 3). These experiments suggest that P2 plays a key stabilizing role in OLF. Perhaps because it participates in a non-covalent interaction network extending to blades B, C, and D, P2 may be a structural element needed for proper native OLF folding, potentially as a component of the folding nucleus or nuclei[44]. Consistent with this notion, the pathogenic substitution K423E is highly destabilizing to OLF[25]. An open question is why P2 was so highly ranked across multiple computational predictions of amyloidogenicity yet does not aggregate as an isolated peptide. This is of particular interest in light of a possible common ancestry between extant functional protein folds and amyloids[45].

In the absence of denaturant or mutation, OLF$^{WT}$ fibrils only grow at 37 °C with a perturbation that enables it to become partially folded. Triggers such as mechanical rocking, low pH, the addition of SDS or peroxide, or an increase in temperature to 42 °C initiate and/or accelerate fibril formation[3,19], similar to reports for lysozyme[30–33], SMA[36], and B2M[34]. However, unlike these model systems, OLF$^{WT}$ exhibits limited ps-µs scale motions typically associated with access to a well-defined low-lying, partially folded excited state that is prone to aggregation[37,46–48]. Characterization of a specific state or ensemble of states by NMR is confounded by the unexpected finding that OLF$^{WT}$ appears trapped in a long-lasting non-equilibrium condition (Fig. 6c). This equilibration rate is well beyond the lifespan of myocilin in eye cells, which is reported to be just a few hours[49], and changes to the protein were not detectable across numerous analyses. It is tempting to suggest that this behavior is necessary in the context of its physiological role in an extracellular eye tissue, to withstand cyclic mechanical stretching associated with blinking or extensive exposure to UV radiation from sunlight. We are unaware of other examples of proteins with this type of long-lived kinetics, especially between apparently well-folded states, and this feature of OLF is of interest for future study.

Even though partial folding is necessary for otherwise globular proteins to begin to fibrillize[50–54], relationships between fibrillization and unfolding are less established. For model amyloid systems, aggregation is sometimes promoted by the formation of a particular folding intermediate on the folding-unfolding pathway[2,35,46,53,55], whereas in other systems[55–57], the relationship between aggregation-primed states and folding-unfolding is less definitive. For OLF, the fibrillization and chemical unfolding pathways are in significant competition at 37 °C (Fig. 7d). Apparent equilibration with urea is rapid, and ThT-positive aggregates accumulate at 37 °C with the addition of just 0.25 M urea. Increasing concentrations of urea leads to the persistent accumulation of partially folded OLF$^{WT}$, and this denaturation correlates with increasing rates of aggregation. At the midpoint of unfolding under equilibrium conditions, where there is the least extent of interconversion between folded and unfolded OLF, aggregation is fastest, indicating that the aggregation-prone species is well-populated and thermodynamically favored. In addition, under ambient conditions, refolding near the midpoint of unfolding is possible, even at relatively high protein concentrations. OLF cannot be refolded and fibrillization is not observed beyond the midpoint of chemical unfolding (Fig. 2). The scenario with OLF is different from light chain variable domain amyloidosis, which requires extensive unfolding to promote fibrillization either by guanidinium HCl or acid[36,58] and in turn, the mature fibril comprises nearly the entire protein chain[59]. Although high urea may stabilize the unfolded state to prevent aggregation, as seen for intrinsically-disordered Aβ[60], NMR spectra show that above the unfolding midpoint, OLF$^{WT}$ adopts a conformationally heterogeneous structural ensemble consistent with a molten globule-type ensemble (Fig. 4a), suggesting that in principle, aggregation could still proceed beyond the midpoint of unfolding. Instead, our interpretation of the data point to the possibility that APRs in OLF cannot be fully unfolded for templated fibrillization to occur. This observation leads to the prediction that unlike the case of light chain amyloidosis, only a

fraction of OLF residues would be contained within the mature fibril, a hypothesis for further testing.

Topological considerations, such as contact order and other allosteric networks like metal centers increase both the complexity of the OLF folding-unfolding pathway, and the likelihood for competition with fibrillization. OLF$^{D380A}$ and OLF$^{I499F}$ adopt differentially non-native structures, indicating that there may not be a common non-native ensemble fingerprint for a pathogenic OLF variant. OLF does not fall neatly into paradigms of well-studied systems that display two-state unfolding and are truly at equilibrium under all desirable experimental conditions. Nevertheless, these OLF studies represent a glimpse into the folding pathways of a larger and more intricate human protein, suggest potential treatments for OLF-specific disease, and provide a vignette into the liabilities that arise from coupled protein folding-unfolding and pathogenic fibrillization pathways.

## Methods

### Expression and purification of OLF for NMR experiments

Plasmids for OLF$^{WT}$, OLF$^{D380A}$, and OLF$^{I499F}$, which encode for an N-terminal maltose binding protein (MBP) and Tobacco Etch Virus protease (TEV) cleavage site before the OLF domain[27] were used for expression in *E. coli* T7 Express cells using the protocol previously reported[23], with minor adjustments. A single colony was selected from a fresh transformation on an ampicillin resistant Luria–Bertaini (LB) broth agar plate, added to 100 mL of LB supplemented with 60 µg/L of ampicillin and grown overnight (37 °C, 225 RPM) in LB. This starter culture was used to grow the cells to an optical density at 600 nm (OD$_{600}$) = 5, at 37 °C in Superior Broth (US Biological Life Sciences) supplemented with 60 µg/L of ampicillin, 500 mL per 2 L flask. At this point cells were pelleted via centrifugation at 4420 × g for 10 min, and washed twice with ~100 mL of 1X M9 salts (26 g/L Na$_2$HPO$_4$ • 7H$_2$O, 3 g/L KH$_2$PO$_4$, 0.5 g/L NaCl, 1 g/L $^{15}$NH4Cl). Washed cells were resuspended in 500 mL minimal media (380 mL sterilized water, 10 mL 20% glucose (sterile filtered), 100 mL 5X M9 salts (sterilized), 0.5 mL 1 M MgSO4 anhydrous (sterile filtered), 0.5 mL 1X Trace Elements Solution (diluted from 1000X sterile filtered solution composed of 40.8 mM CaCl$_2$ • 2H$_2$O, 21.6 mM FeSO$_4$ • 7H$_2$O, 6.1 mM MnCl$_2$ • 4H$_2$O, 3.4 mM CoCl$_2$ • 6H$_2$O, 2.4 mM ZnSO$_4$ • 7H$_2$O, 1.8 mM CuCl$_2$ • 2H$_2$O, 0.3 mM boric acid, 0.2 mM ammonium molybdate, 5 g/L EDTA), 0.1 mL 1 M CaCl$_2$ (sterile filtered) and 2.5 mL BME vitamins (Sigma Aldrich, B6891-100ML, sterile)). Resuspended cells were incubated for 2 h (37 °C, 225 RPM), cooled to 18 °C and induced with 1 mM IPTG prepared in water, and allowed to grow for 48 h. Cells were harvested by centrifugation (4420 x *g*, 4 °C, 10 min), and pellets were flash frozen with liquid nitrogen and stored at −80 °C. For the uniformly double ($^{13}$C, $^{15}$N) labeled sample, 20 mL of 20% $^{13}$C glucose per liter of media (sterile filtered) was used as the carbon source. This amount was doubled to 40 mL of 20% glucose per liter for $^{15}$N single labeled samples. Cells were lysed and OLF was purified as previously described[27]. Specifically, cell pellets were lysed via French Press in 10 mM KH$_2$PO$_4$, 10 mM Na$_2$HPO$_4$, 200 mM NaCl, 1 mM EDTA (amylose wash buffer), and supplemented with a Roche Complete Protease Inhibitor Cocktail tablet. The lysate was clarified via centrifugation (110,000 x *g* for 1 h), and loaded onto a 20 mL amylose resin column equilibrated with amylose wash buffer. The MBP-OLF fusion protein was eluted with amylose wash buffer supplemented with 10 mM maltose, concentrated using Amicon Ultra centrifugation devices with a 30 kDa cutoff, and loaded onto a gel filtration Superdex 75 prep-grade column equilibrated with 10 mM KH$_2$PO$_4$, 10 mM Na$_2$HPO$_4$, 200 mM NaCl (phosphate gel filtration buffer, PGF), pH 6.8. Fractions corresponding to monomer MBP-OLF were pooled and subjected to cleavage with TEV protease at room temperature overnight. TEV protease was produced in-house. After cleavage, TEV protease was removed via nickel affinity using a 5 mL HisTrap, followed by an amylose affinity column to remove MBP. Any aggregates were removed from the OLF solution via a second

Superdex 75 column and the purification was completed with a final amylose column where OLF eluted in the flowthrough fraction. Purity was verified via SDS-PAGE. For 2D NMR experiments, [15]N-labeled OLF was concentrated to ~50 µM in PGF pH 6.8 using Amicon Ultra centrifugation devices (molecular weight cutoff 10 K) and shipped to ASU overnight on ice packs. For 3D NMR experiments, [13]C, [15]N-labeled OLF was concentrated to ~500 µM in PGF. Concentrations were calculated using the calculated extinction coefficient for OLF 68,425 $M^{-1}cm^{-1}$ from ExPasy ProtParam (MW [15]N OLF WT = 31,518 g/mol, MW [15]N OLF[D380A] = 31,474 g/mol, MW [15]N OLF[I499F] = 31,552 g/mol, MW [13]C/[15]N OLF WT = 32,918 g/mol)[61]. Average yields from 1 L minimal media were 0.04 mg/L for OLF[D380A], 0.02 mg/L for OLF[I499F] and 2.2 mg/L for WT.

## Metal analysis
As-purified OLF[WT] and OLF[I499F] were submitted for inductively coupled plasma-optical emission spectroscopy analysis (University of Georgia Center for Applied Isotope Studies). Freshly purified samples of ~30 µM protein and buffer blanks were analyzed for $Ca^{2+}$ content. Data were blank-subtracted; molar ratios were calculated and reported. Concentrations of OLF[WT] and OLF[I499F] were calculated by Beer's law as described above.

## NMR spectroscopy and backbone resonance assignments
All NMR spectra were collected on a Bruker 850 MHz [1]H spectrometer and Avance III HD console equipped with 5 mm TCI Cryoprobe. NMR sample temperatures were calibrated using standard 99% [2]H methanol[62] to within ~0.1 °C. NMR data were collected with Topspin 3.6.2

All NMR experiments used for backbone resonance assignment were collected at 25 °C using a 3 mm NMR tube with 2.2% $D_2O$ (4 µL in 160 µL total volume). [13]C, [15]N WT-OLF sample was prepared at 570 µM in PGF, pH 6.8. All 2D and 3D experiments were TROSY based. The 3D experiments used for backbone assignments include HNCA, HNcoCA, HNCACB, CBCAcoNH, HNCO, and HNcaCO. The 3D data were collected using non-uniform sampling and Poisson gap sampling schedules[63]. Because of concerns about protein stability, most 3D experiments were recorded two (HNCA, CACAcoNH, HNCO) or three (HNcoCA, HNCACB) times. The HNcaCO experiment was collected once. The resulting spectra were reconstructed using an iterative soft threshold, processed, and added together using NMRPipe[64] prior to analysis. Sample integrity was monitored by [1]H-[15]N TROSY-HSQC prior to and following each 3D experiment. Supplementary Table 7 details NMR experimental parameters for backbone assignment experiments. Backbone resonance assignments relied on CcpNmr AnalysisAssign V3 (v3.0.4)[65,66]. After peak picking in CcpNmr, initial tentative assignments were generated with the i-Pine server[67], followed by traditional manual resonance assignment. We note that some resonances have shoulders, which could indicate slow time-scale dynamics. However, the current level of assignments and data quality do not allow for confident assessment of the origins of the less intense resonances that are overlapping on some more intense resonances.

## NMR-detected OLF[WT] urea titration
A 300 µM stock of [15]N OLF WT in PGF and a 9 M stock solution of urea in water were used to prepare 50 µM WT-OLF samples in six different urea concentrations (0 M, 1 M, 2 M, 3 M, 4 M, 6 M). These samples were mixed in microcentrifuge tubes to a total volume of 175 µL in PGF, pH 6.8 buffer, incubated for 1 h at room temperature, and added to 3 mm NMR tubes with 2.8 % $D_2O$ (5 µL). [1]H-[15]N TROSY-HSQC spectra were collected at 25 °C. All urea samples were incubated on ice for 1 h immediately prior to data collection. The resulting NMR data were processed using NMRPipe and analyzed using principal component analysis (PCA) within the TRENDNMR software package[68]. TRENDNMR output four principal components (PC1-PC4), the first principal component (PC1; 29% contribution) of each urea titration point was fit to a two-state model to obtain the urea concentration at the midpoint of unfolding ($C_m$):

$$PC1 = PC1_{min} + \frac{(PC1_{max} - PC1_{min})}{1 + 10^{((\log(C_m) - [Urea]) \cdot n)}} \quad (1)$$

where n is the Hill coefficient. Additional experimental parameters for the NMR-detected urea titration and other 2D data detailed below are listed in Supplementary Table 8.

## NMR characterization of OLF[D380A] and OLF[I499F]
OLF[D380A] and OLF[I499F] were prepared in 3 mm tube with 2.8% $D_2O$ (5 µL; 180 µL total volume) in PGF, pH 6.8. [1]H-[15]N TROSY-HSQC spectra were collected at 25 °C. Spectra were processed with NMRPipe.

## CSP analysis of OLF [1]H-[15]N TROSY-HSQC spectra
Chemical shift perturbation (CSP) was evaluated using CcpNmr AnalysisAssign V3. The assigned resonances for OLF[WT] were mapped to corresponding spectra for mutants OLF[D380A] and OLF[I499F] and 1 M and 2 M urea. In an effort to minimize user bias, the OLF[WT] assignment peak list was used as a reference target, and the mutant/urea spectral peak lists were treated as globally mobile and fit to the reference target data. The global spectral differences were minimized using a sum of Euclidean distances algorithm within MATLAB R2022a. Based on the spectral minimization, assignments were then mapped onto the closest resonance from each mutant/urea spectrum. Each transferred assignment was then manually inspected. In an effort to minimize potential overinterpretation of the data, in cases of significant peak discrepancies, resonance assignments were not propagated to the mutant/urea spectra. Generally, only resonances that were partially overlapped or in unique spectral regions (e.g. far distant from others) were assigned. Due to poor resonance dispersion in the center of the OLF[I499F] spectra, peaks between 113-128 ppm in the nitrogen dimension and 7.9-8.6 ppm in the hydrogen dimension were unassigned. Similarly, because of the collapse of resonance dispersion in OLF high urea spectra, only 1 and 2 M urea spectra were analyzed and CSP quantified.

Quantification of changes in chemical shift position (Δδ) relative to OLF[WT] were carried out using the following equation:

$$\Delta\delta = \sqrt{(\Delta\delta_H)^2 + (0.2(\Delta\delta_N))^2} \quad (2)$$

Δδ values were visualized on the OLF structure (PDB 4WXQ, [https://doi.org/10.2210/pdb4WXQ/pdb]) in PyMOL software. Unassigned WT peaks were shaded gray. Assigned WT peaks not present, as described above, in the mutant/urea spectra were colored black. Variable CSP was represented by color gradients on the OLF structure from blue to red (0-0.5 Δδ ppm).

## OLF[WT] aggregation kinetics
A ThT stock solution (500 µM ThT in deionized water) was diluted to a 200 µM working solution in PGF, pH 7.2 for aggregation experiments. OLF (30 µM) in PGF buffer, pH 7.2 was incubated with a range of urea concentrations (0, 0.25 M, 0.5 M, 0.75 M, 1 M, 1.25 M, 1.5 M, 1.75 M, 2 M, 2.5 M, 3 M and 6 M) plus 10 µM ThT. OLF samples with urea and ThT were prepared at a final volume of 150 µL in triplicate in microcentrifuge tubes, mixed via pipetting and then transferred to a 96-well microplate (Grenier). Plates were sealed with clear MicroAmp PCR film (Applied Biosystems) and placed in a Biotek Synergy microplate reader equipped with a 440 nm excitation filter and a 485 nm emission filter. Fluorescence measurements were recorded every 10 m at 37 °C and background subtracted with a PGF, pH 7.2 control supplemented with 10 µM ThT. After 96 h, the samples were mixed thoroughly in the well before removing the solution to a microcentrifuge tube and then centrifuged on a tabletop centrifuge at 11,600 x·g and 4 °C for 10 min so that the pellet could be inspected for yellow tint associated with

ThT binding. ThT fluorescence was averaged for each sample in triplicate and then plotted as a function of time using GraphPad Prism software.

## OLF[WT] unfolding and refolding measured by CD

To assess tertiary structure of OLF with different concentrations of urea, near-UV CD was performed as reported previously[19,24]. Experiments were conducted with OLF in PGF, pH 7.2 at a final concentration of 30–50 μM on a Jasco J-810 polarimeter. For unfolding, OLF was allowed to equilibrate in 3 M urea for at least 1 h at 4 °C prior to acquisition of CD spectra. For refolding, OLF was equilibrated in either 3 or 5 M urea for at least 15 m at room temperature before being diluted 10-fold with PGF buffer. Spectra were recorded within 20 m of diluting with PGF. Spectra were acquired at 4 °C with 10 averaged scans from 320 to 250 nm at a 50 nm min$^{-1}$ scan rate, using a 0.1 cm cuvette. Data were blank-subtracted and converted to mean residue ellipticity

$$\theta = \frac{M_{res} \cdot \theta_{obs}}{10 \cdot d \cdot c} \qquad (3)$$

where $M_{res} = 112.9$ is the mean residue mass calculated for OLF; $\theta_{obs}$ is the observed ellipticity (degrees) at a given wavelength; d is the path length (cm); and c is the protein concentration (g/mL). Data is representative of 2 independent protein preparations for each urea concentration.

## OLF[WT] unfolding measured by fluorescence

OLF[WT] that had been flash frozen after purification (2 protein preparations, one in analytical duplicate, and one in analytical triplicate) or OLF[WT] that had been stored at 4 °C for 1 month (stored, one protein preparation, in analytical triplicate) was used for experiments. Buffer, 7 M urea, and fresh or stored OLF[WT] were equilibrated at either 25 °C or 37 °C for at least 30 min prior to sample preparation. Samples were prepared by diluting to a final concentration of 1 μM OLF in a final concentration of 0 – 6 M urea and then equilibrated at the respective temperature for an additional 1 h or 48 h prior to fluorescence measurements. Fluorescence spectra were measured at the corresponding temperature (25 °C or 37 °C) on a Horiba FluoroMax spectrofluorometer equipped with a Peltier temperature control system. Emission spectra were measured from 300–400 nm ± 5 nm at 1 nm intervals with an excitation wavelength of 295 ± 1 nm. Each recorded wavelength emission maximum($\lambda_{max}$) value is the average of 5 scans on the same sample. Spectra were first smoothed according to a 2$^{nd}$-order polynomial averaging 5 neighbors. The emission wavelength corresponding to the highest fluorescence intensity was recorded as $\lambda_{max}$. Urea unfolding as a function of $\lambda_{max}$ was fit to a Boltzmann sigmoidal function to determine the halfway point of unfolding. Data smoothing and analysis was performed in GraphPad Prism v9.

## HDX-MS

HDX-MS was performed on OLF[WT], OLF[D380A], and OLF[I499F] on three separate days, within a two month period, using a Waters HDX system with nanoAcquity UPLC and Micromass Q-ToF Premier mass spectrometer (Waters Corp, Milford, MA) in 3 replicates. 30 μM of purified protein in PBS was mixed with D$_2$O-containing buffer (10 mM phosphate buffer, 99.9% D$_2$O, pD 7.0) at 1:7 (v:v) ratio by an automated LEAP pipetting robot for 10 to 10 000 s at 20 °C. OLF was mixed with H$_2$O-containing buffer for the 0 s time point. Each time point was repeated 5 times. The deuterium exchange reaction was quenched with an equal volume of precooled buffer containing 100 mM phosphate, 0.5 M tris(2-carboxyethyl)phosphine, 0.8% formic acid, 2% acetonitrile, pH 2.5 for 180 s at 1 °C. The quenched reaction was digested on a Waters Enzymate BEH Pepsin Column (2.1 × 30 mm) at 20 μL min$^{-1}$. Peptide fragments were separated on a Waters ACQUITY UPLC BEH C18 column (1.7 μm, 1.0 × 100 mm, 40% - 90% acetonitrile

gradient) at 40 μL min$^{-1}$ and 1 °C. Mass spectrometry was performed with the electrospray ionization source in positive ion mode. A lock-mass of Glu-Fibrinopeptide (Sigma-Aldrich, St Louis, MO) was collected together with each sample as an internal reference. Peptides were identified using the OLF sequence in Protein Lynx Global Software (ver.3.0.2). HDX-MS data was analyzed manually using the DynamX (ver.3.0) software with removal of any fragments with >0.2-Da mass deviation (Supp. Table S3).

## MD simulations

The initial structure of OLF[WT] was taken from PDB 4WXS [https://doi.org/10.2210/pdb4WXS/pdb][17]. A disulfide bond between residues C245 and C433 was added. Standard protonation states were assumed for all other residues. The Ca$^{2+}$ and Na$^+$ ions present in the structure, as well as all crystallized water molecules, were retained. The resulting system was solvated with 21041 water molecules, and 150 mM NaCl was added to the bulk water using VMD[69], giving a system size of 68,238 atoms. The OLF[D380A] and OLF[I499F] mutants were created by mutating the appropriate residue with Psfgen in VMD. Each system was equilibrated in stages using the simulation software NAMD3[70] and the CHARMM36m protein force field[71]. First, all protein atoms and crystallized ions were restrained for 0.5 ns, followed by releasing the side chains for 1.5 ns. Next, 10-μs production simulations in triplicate for each system were run using Amber18[72]. A time step of 2 fs was used for equilibration, while 4 fs with hydrogen mass repartitioning[73] was used for production runs. Long-range electrostatic interactions were evaluated using the particle-mesh Ewald method[74]. A cutoff of 12 Å was used for van der Waals interactions. A constant temperature of 293 K (20 °C) was maintained using Langevin dynamics, while a constant pressure of 1 atm was enforced using the Langevin piston method for the initial 2 ns of equilibration to ensure the volume was stable; all three systems stabilized at a volume of (87.3–87.4 Å)$^3$.

## Crystallization, data collection, and structure refinement

Purification of OLF[WT] was completed 1 day prior to setting up crystallization drops in a 24-well hanging drop crystallization tray (Hampton Research). OLF[WT] was buffer exchanged into 10 mM HEPES pH 7.2, 0.2 M NaCl and concentrated to 10 mg mL$^{-1}$ using an Amicon filtration unit with a 10 kDa molecular-weight cutoff. The reservoir solution of 500 μL was composed of 5% PEG 8000 and 0.05 M magnesium formate. A hanging drop of 4 μL was prepared on a plastic cover slip at 1:1 (v:v) ratio of OLF[WT] and the reservoir solution. Crystals were observed after 4 days of incubation at 16 °C. After ~10 months, crystals were harvested and cryoprotected by incubation for 2.5 h in the reservoir solution supplemented with 3% glycerol and 0.1 M azetidine-3-ol, followed by flash-cooling in liquid nitrogen. Diffraction data was collected on the Southeast Regional Collaborative Access Team (SER-CAT) 22-ID beamline at the Advanced Photon Source (APS) using SER-GUI and were processed using HKL-2000[75]. Molecular replacement was performed with *Phaser*[76] using the polypeptide chain of WT myocilin OLF (PDB entry 4WXQ [https://doi.org/10.2210/pdb4WXQ/pdb][17]) as a search model with noncovalently bonded molecules removed. The model was built and refined using *Coot*[77] and phenix.refine[78]. For multiconformer analysis and refinement, qfit-3.0 was used[79]. A composite omit map was generated in Phenix, qfit was run using a step size of 40 degrees, range/neighborhood sampling 45 degrees, and disabling parsimonious selection of the number of output conformers, and the output multiconformer model was refined using *phenix.refine*. Figures were prepared in PyMOL. The structure was deposited in the PDB with accession code 8FRR [https://doi.org/10.2210/pdb8FRR/pdb].

## LC-MS/MS analysis for posttranslational modifications

Protein samples (fresh OLF[WT] and OLF[WT] used for NMR stored at 4 °C for ~1 year) were denatured by incubating in 6 M urea, 50 mM Tris-HCl

buffer (pH 8.2) at room temperature for 30 min. Denatured samples were reduced with 5 mM dithiothreitol (DTT) in dark at 56 °C for 25 min, cool down to room temperature, and alkylated with 14 mM iodoacetamide (IAA) in dark at room temperature for 30 min. Chymotrypsin was added at a final protease: protein ratio of 1:30 (w/w) and incubated with samples at 25 °C for 18 h. Digested samples were desalted through Sep-Pak tC18 cartridges and dried by CentriVap. Reconstituted samples were analyzed by LC-MS/MS using a Q-Exactive Plus Orbitrap mass spectrometer equipped with Dionex UltiMate 3000 LC system. Briefly, peptides were resuspended in 10% acetonitrile/ 0.1% formic acid and loaded onto a trap column (PepMap™ NEO 5 μm C18 300 μm X 5 mm Trap Cartridge) and resolved through a custom analytical column packed with ReproSil-Pur 120 C18-AQ 3 μm beads (Dr. Maisch GmbH) at a flow rate of 0.3 mL/min with a gradient solvent A (0.1% formic acid in 2% acetonitrile) and a gradient solvent B (0.1% formic acid in 80% acetonitrile) for 150 min. MS analysis was conducted in a data-dependent manner with full scans in the range from 400 to 1800 m/z using an Orbitrap mass analyzer set as follows: MS1: resolution = 70,000, AGC target = $3e^6$, Max IT = 100 ms; MS2: resolution = 17,500, AGC target $1e^5$, Max IT = 50 ms. The top fifteen most intense precursor ions were selected for MS2 with an isolation window of 4.0 m/z. Isolated precursors were fragmented by higher energy collisional dissociation (HCD) with normalized collision energy (NCE) of 27. Three analytical runs were conducted for the fresh OLF[WT] sample and the NMR sample.

LC-MS RAW files were searched against the C-terminal olfactomedin domain of myocilin sequence using the Sequest HT search engine embedded in Proteome Discoverer 3.0 with 10 ppm MS1 precursor mass tolerance, 0.02 Da MS2 fragment mass tolerance, 0.01 false discovery rate. For the fresh sample, oxidation (M) and deamidation (N, Q) were set as dynamic modifications, carbamidomethylation was set as static modification. For the NMR sample, the results combined [15]N labeled and unlabeled analyses. The [15]N unlabeled analysis was set the same as the fresh sample; for the [15]N labeled analysis, [15]N labels on C, M, N, Q together with oxidation (M), deamidation (N, Q) and carbamidomethylation (C) were set as dynamic modifications and [15]N labels on the other amino acids were set as static modifications. The modified PSM% of a modification site was calculated as the number of modified PSMs divided by the number of total PSMs containing this site. Modified PSMs number is the number of PSMs where the modification site has the modification (site localization probability higher than 90%). Total PSMs number is the number of PSMs that contain the modification site (modified and unmodified). The modified PSM% was calculated within each technical replicate and the mean modified PSM % with a standard deviation was compared between the same modification sites detected in the old NMR sample and the fresh sample.

## Reporting summary
Further information on research design is available in the Nature Portfolio Reporting Summary linked to this article.

## Data availability
The NMR data generated in this study have been deposited in the BMRB database under accession code 51750. The 1.27Å resolution OLFWT crystal structure data have been deposited to the PDB under accession code 8FRR. Previous OLF[WT] crystal structures have been deposited to the PDB under accessions 4WXQ and 4QXS. The HDX-MS data have been deposited to the ProteomeXchange Consortium with the dataset identifier PXD045520. All other data generated in this study are provided in the Supplementary Information/Source Data file. Source data are provided with this paper.

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

## Acknowledgements

Research reported in this publication was supported by the National Institutes of Health award numbers R01EY021205 (RLL, WVH), R41EY031203 (RLL), R01GM123169 (JCG), and R35GM141933 (WVH). EGS, HFS, and MTM were supported in part by 5T32EY007092-35. We thank members of the Renhao Li lab (Emory) for assistance with HDX-MS, Brett Barlow for technical assistance with X-ray data collection, and Dr. David Smalley for assistance with mass spectrometry. We acknowledge the Petit Institute for Bioengineering and Biosciences at Georgia Institute of Technology, the Magnetic Resonance Research Center at Arizona State University, and the Robert P. Apkarian Integrated Electron Microscopy Core (IEMC) at Emory University for access to core facilities. The IEMC is subsidized by the School of Medicine and Emory College of Arts and Sciences and Georgia Clinical & Translational Science Alliance of the National Institutes of Health under award number UL1TR000454. Synchrotron data were collected at Southeast Regional Collaborative Access Team (SER-CAT) 22-ID beamline at the Advanced Photon Source, Argonne National Laboratory. SER-CAT is supported by its member institutions, and equipment grants (S10_RR25528, S10_RR028976 and S10_OD027000) from the National Institutes of Health. Use of the Advanced Photon Source was supported by the U. S. Department of Energy, Office of Science, Office of Basic Energy Sciences, under Contract No. W-31-109-Eng-38. The content is solely the responsibility of the authors and does not necessarily represent the official views of the National Institutes of Health.

## Author contributions

W.D.V-H. and R.L.L. conceived of the study. E.G.S. and S.E.H. prepared samples for NMR. M.D.M., M.K., E.G.S. conducted NMR experiments. E.G.S. and H.F.S. conducted unfolding and aggregation assays. M.T.M. prepared samples for HDX-MS. M.T.M. and R.L. analyzed and interpreted HDX-MS data. E.R. and J.C.G. conducted and interpreted MD simulations. X.S. and M.P.T. conducted and analyzed MS data. E.G.S., M.D.M., W.D.V-H, R.L.L. analyzed data and wrote the manuscript. M.T.M., X.S., R.L., M.P.T., and J.C.G. edited the manuscript. All authors have given approval to the final version of the manuscript.

## Competing interests

The authors declare no competing interests.
