## [Peer Review File · Nature Communications]

Competition between inside-out unfolding and pathogenic aggregation in an amyloid-forming β -propellerREVIEWER COMMENTS

Reviewer #1 (Remarks to the Author):

The problem investigated in this paper is a very interesting and important one, namely, how mutations in the olfactomedin (OLF) domain of myocilin may cause glaucoma by promoting misfolding and amyloid aggregation. The authors report characterization of the unfolding and aggregation of wild-type OLF in denaturant (urea) and of moderate (D380A) and severe (I499F) OLF glaucoma variants using a suite of complementary structural and biophysical methods. The current understanding of molecular details of aggregation for OLF is very limited; understanding thereof is significant also for gaining insight into how other relatively larger and more complex protein folds behave. Based on NMR chemical shift perturbations and molecular dynamics simulations (MD), the 5-blade toroidal structure of OLF is proposed to unfold from the inside out, starting with loss of a Ca ion bound in the center of the structure, followed by propagation of unfolding towards the outer b-strands, which retain structure the longest. Based on crystallography, MD, and NMR, OLF may undergo complex dynamics on timescales ranging from ps to weeks, suggesting it is not at equilibrium and may form intermediates in competing unfolding and aggregation pathways.

The authors describe a breadth of detailed results that may help unravel the still obscure complexities of how gain-of-function mutations promote toxic misfolding and aggregation in disease. The differences for D380A and I499F are particularly noteworthy. There are multiple points that need to be addressed to further define and validate the results and their interpretation.

Important points:

1. The proposed model for structure disruption is that Ca is lost first; this step needs to be further validated. There is evidence that Ca binding is diminished in D380A, consistent with mutation of this key Ca-liganding residue. Could there also be perturbations to sodium binding that impact the protein behaviour? Are WT and I499F fully metal bound? Additional information is needed on the effect of urea on metal binding, for WT, D380A, and I499F. For example, a denaturation curve monitored by CD and/or NMR, with more urea concentrations than in Figure 4 (and/or for different metal concentrations), might show evidence for multiple transitions, i.e. loss of metal binding followed by denaturation. It is important to include urea denaturation data for the mutants (as for WT in Figure 2D,C).
2. The conclusion that amyloid aggregation competes with chemical unfolding but only below the threshold where OLF loses discrete tertiary structure needs clarification and further consideration. Since urea and mutations promote aggregation under conditions where the protein is presumably predominantly folded, how does this competition occur? Also, the prominent random coil signals in the spectrum of I499F (Figure 5) could be due to disruption of the monomer structure, as proposed, or be caused by protein self-association. Are the spectra of the various proteins constant with time or varying due to changing proportions of monomer and self-associated species?

3. The effects on protein structure and amyloid aggregation of D380A are proposed to resemble those of 1 M urea on WT, while the I499F effects resemble those of 2 M urea on WT. The authors hypothesize that OLF fibrillization proceeds via access to partially unfolded states. It may be that multiple partially unfolded states are promoted by the toroidal fold of OLF, which might initially unfold not just near the metal binding site but also in the region where the N and C termini associate, in a process that is less cooperative than for typical globular proteins. I recommend adding some discussion of these points to emphasize their significance.

4. The minor changes observed by NMR for WT over months might arise from various covalent changes, e.g. deamidation, oxidation, cleavage not apparent by SDS-PAGE. It is important to check for such modifications e.g. by mass spectrometry.

Minor points

1. Please add numbers for the x axis to Figure 1A, and add explanation for the blue spheres in the legend for Figure 6C. I think A and B labels are swapped in Figure 7; please specify whether teal indicates no significant CSP and the quantitative criteria for yellow or teal color. Also, include explanation of color for bound metals.

2. Many resonances for WT in Figure 3 may exhibit splitting (e.g. shoulders on T259, S393, I432, D454, maybe nearby minor peaks for G266), or these unlabelled minor signals may arise from residues that are not assigned. Please elaborate on this aspect (e.g. in the SI).

3. Please include additional quantitative details for the principal component analysis of Figure 4 e.g. how many significant components were identified?

4. RMSF values are not shown for the (most mobile) N-terminal residues in Supp. Fig. S6. Please add information on these values (if too large, can state in the legend).

Reviewer #2 (Remarks to the Author):

The authors report molecular insights into the fibrillogenesis of the protein OLF and two variants. Their study shows that fibril formation likely proceeds through the population of an intermediate conformation with oligomerisation ablated at high urea concentrations where the fully unfolded state dominates the folding landscape. The studies are complimented with a considerable amount of NMR which further reveals an inside-out unfolding transition preceding fibre formation.

The authors have performed a lot of experiments and while I am no expert in the area in general I agree with the main conclusions but I have some of my own points/comments.

I have been specifically asked by the editor to look at the HDX-MS aspects of the m/s so I'll start here. The authors are evaluating isotope incorporation directly and not using the more conventional approach comparing the HDX behaviour of one protein by comparison to a reference sample. There is nothing wrong with this approach but I note that the authors have not taken any control measurements to

account for back and forward exchange artifacts. Back exchange artefacts in particular can be considerable and lead to false interpretation of experimental data if they haven't been accounted for. As things stand they cannot use the current data and if they want to include HDX-MS dynamics in the article they will have to perform new acquisitions and they have 2 options.

First, reacquire the experimental data but with 3 replicates of back exchange controls and 3 replicates for forward exchange controls. Back exchange control samples are fully in-exchanged protein (the same as the experimental protein) acquired as a 15 second labelling experiment. In exchange can be performed by unfolding the protein in excess D2O and then refolding by dialysis against D2O or some equivalent using diafiltration. Alternatively the protein can be incubated in labelling buffer for several days at 37 degrees C this normally exchanges all the 1H for 2H. Ideally aliquots would be taken at several timepoints during incubation to see when the D-incorporation has plateaued but 7-days incubation is normally sufficient. Forward exchange controls constitute a reference acquisition on 1H protein but with the quench made in D2O so that the final D2O concentration in the quench is 50%. Correction of the RFU for each timepoint and each peptide is then made using the following: $RFU_{corr} = (RFU_{obs.} - RFU_{wdexch.}) / (RFU_{backexch.} - RFU_{wdexch.})$

Second, the authors could use the more conventional difference data approach and in their case obtain experimental data for the WT and one of the variants and report deltaRFU. Note for this they would still have to take back exchange control data and correct the RFU, forward exchange correction is considered unnecessary for this type of data.

Note the acquisition of forward and back exchange control data cannot be taken after the fact. The control data must be taken in the same window as the experimental acquisitions.

Further comments

F2 A change colour key remove [urea] line

F2 C The test suggests that the differences between 3 and 5 M urea are due to competition between refolding and aggregation and at 5 M aggregation wins. But at higher urea concentrations could the protein just be kinetically trapped in a higher energy unfolded state? It's probably not, but the authors should indicate refolding time so we can be sure the system is in a steady state. Also find it odd that the protein cannot refold. Once chaotrope is removed the protein is no longer unfolded or folded where is it the authors didn't state, does it precipitate or form fibres? – clarify.

F3 B can the authors indicate on the figure the residues missing from the xtal structure for quick reference.

Line 208 "This result emphasizes the central importance of the Ca²⁺ site in the structural integrity of the OLF propeller." What do the authors mean by this point can they clarify? Taken as read they mean the site not the ion and that loss of structural integrity at the site perturbs that whole structure. If ion is important in structural integrity the authors should demonstrate this with wild-type but with the addition of a chelator to remove the calcium.

Figure 7d is confusing. There should be 2 folding landscapes one for monomers and one for different types of multimers with both landscapes being linked by a common monomeric intermediate. The current image doesn't accurately represent the authors claim that the fibril form is accessed via an intermediate in the current representation the fibril can be access from the fully unfolded state which is at odds with the experimental data. The fibril is also lower energy that the native monomer which I don't think is necessary true. 2 landscapes joined by a common intermediate would be much clearer.

Reviewer #3 (Remarks to the Author):

Saccuzzo et al. analyzed the structural dynamics and instabilities of the olfactomedin (OLF) domain of myocilin. Mutations in this 5-bladed beta-propeller protein are causal for early-onset hereditary open-angle glaucoma, which is connected to amyloid aggregation of OLF. It is therefore of upmost interest to understand the protein dynamics that leads from the folded protein to the amyloid state. To this end, the authors performed a number of NMR spectroscopy experiments as well as MD simulations to resolve the protein dynamics under different conditions (causing protein unfolding at increasing urea concentrations and by introducing disease-relevant point mutations) and at different time scales. The most important findings are that structural changes leading to unfolding start at the inside of the beta-propeller, which are encouraged by loss of the internal Ca²⁺ center and propagate outward to the surface of the propeller. However, complete unfolding as provoked by very high urea concentration do not lead to amyloid fibrillization.

This is an interesting study. The most exciting finding is that the unfolding starts in the interior and not exterior of the beta-propeller.

Based on my solid knowledge of NMR spectroscopy (though I am not an expert in it), I would say that the NMR experiments (and also H-D exchange mass spectrometry) were performed very carefully and at a high level of standards.

However, the MD simulations are below standard in terms of quantity and approach. First, the simulations should be repeated at conditions mimicking the different experimental conditions by adding urea to the WT and introducing mutations D380A and I499A. Moreover, either much longer (10 microseconds) or enhanced simulations (REMD) should be conducted to observe structural changes that the could be related to the experimental observations. I highly recommend to use another MD engine than NAMD as NAMD is too slow for systems below 500,000 atoms. Gromacs or Amber would be better MD codes for this purpose.

Because of the rather short MD simulations, the results on the protein dynamics presented in Fig. 6 are the weakest part of the study, as there is a missing link between the short- and long-time scale dynamics. The aim (for a manuscript in Nat. Commun.) should be to provide a structural picture that shows how the unfolding starts.

Apart from this, I have some more specific comments:

1) l. 71: Introduce ER (even though it is a common abbreviation)

2) Fig. 2:

i) The color scale is not optimal as the different shades of blue are difficult to distinguish. Please change.

ii) Based on Fig. 2B the authors conclude that at high urea concentrations complete unfolding wins and inhibits amyloid aggregation. It is therefore speculated that the latter is initiated from partially folded

OLF. However, for Abeta - which is never folded in solution at pH ~7 - it is known that high urea concentrations inhibit amyloid aggregation (see, e.g., Protein Sci. 2004 Nov; 13(11): 2888–2898). If one understands amyloid aggregation as a metafolding problem, it is understandable that this does not take place at high urea concentrations. As long as the structure that finally leads to amyloid aggregation is not fully determined, I find the authors' conclusions with regard to their observations at high urea concentrations not justified.

3) Fig. 3 needs to be improved. The residue labels in panel A are not or hardly readable. The size of the figure can be increased to make full use of the page width. The resolution of the figures needs to be increased. And I am sure that also the size of the residue labels could be increased by 1 pt.

4) Fig. 4: The meaning of the residues colored in gray in panel C needs to be explained. Please add spheres for Na⁺ and Ca²⁺ to the structures and letters "A" to "E" at the outside of the blades.

5) Fig. 5: Please add spheres for Na⁺ and Ca²⁺ to the structures, mark the respective mutation site by a star (or alike), and add letters "A" to "E" at the outside of the blades.

6) p. 13: Please add in % how many of the chemical shifts remain almost unchanged upon mutation (and how many cannot be determined).

7) Fig. 6: The same kind of results should be provided for selected urea concentrations and the mutants. This would better allow to link protein dynamics with unfolding and amyloid aggregation. See also my comment with regard to running further MD simulations.

Reviewer #4 (Remarks to the Author):

This paper from the Lieberman lab reports Competition between inside-out unfolding and pathogenic aggregation in an amyloid forming β -propeller. There are several strengths to the work, including unfolding events, that required for amyloid formations. Authors used several techniques including NMR, X-rays, and molecular dynamics simulations to give insight on the OLF domain folding.

Major comment:

- My major concern regarding the novel of this work. The Crystal Structure of the Myocilin Olfactomedin Domain has been solved (PDB: 4WXQ). It will be more interesting if the authors can determine a novel structure of an intermediated species such as Oligomers or final aggregated form such as amyloid fibril structure of OLF domain.
- I am wondering if OLF domain is fibrilize in human patient. Do the authors have any evidence that OLF present in human patients as fibril species?

The authors should address the following comments and recommendations on additional critical experiments to make for a stronger study.

Minor comments:

- The abstract is long, need to be shortened

Introduction

- Missing reference at the end of this sentence. A recent addition to the list of pathogenic proteins that form amyloid is the myocilin olfactomedin (OLF) domain.
- Missing reference at the end of this sentence. Many different single point mutations, dispersed throughout the sequence, are the strongest genetic link to early onset open angle glaucoma.
- Remove figure 1 from introduction, reference is enough. Figure 1, it is not structure solved in this paper and cannot be in the introduction.
- At the end of introduction, Authors required to bring the message of the question need to be answered in this paper (this is not clear here).

Results:

- Authors reported that they monitored kinetics of amyloid formation in the presence of varying concentrations of urea (Figure 2A, S1B). It will be interested to see under Electron Microscope using negative stain, the aggregation products (Oligomers, fibrils, mature fibrils and amorphous aggregates).
- In Figure 1S, it looks like tube with 0 urea, has some aggregation as well. Can author explain?
- In Figure 2C, CD spectra of the untreated sample doesn't seem folded well. I am wondering that the protein is not completely pure or folded after purification. Authors need to show CD spectrum starting from 190 nm, possible there are more interesting spectrum to show.
- The ThT signal, doesn't confirm that the protein can form amyloid fibrils, many ThT signal can also produce from amorphous protein aggregation. Authors should use EM to prove that the OLF can form amyloid fibrils.
- Authors reported at urea concentrations >2 M, the unfolded state is dominant and limits the ability to confidently map assignments and track CSP. I would suggest that Authors need to examine the conformation state under EM first, and see if these conformations are stable structure, they can dialysis the protein against the suitable buffer and recorded the data. It is also possible that at urea

concentrations >2 M, the OLF generate number of heterogenous conformations that limit the assignments.

- Authors reported that this observation may explain why D380A forms P3-like fibrils: if P3 residues are mobile and 214 disordered and not sequestered as in WT, they are available to template fibril formation. There is no experimental evidence that show that D380A forms can form fibrils.
- Authors reported that they solved a 1.27 Å crystal structure of OLF (Figure 6A, Supp. Table S3). The new structure shares overall features with our previously reported ~2 Å resolution structures. The resolution is higher and better but still similar to previous structure, 2 Å structure. I feel that there no need to solve previously published structure. It will be more interested to solve the same structure in the presence of ligand or inhibitors.
- Authors claim that OLF fibrilization in many parts in the paper, especially in the discussion section (For OLF, 439 the fibrillization and chemical unfolding pathways are in significant competition at 37 °C 440 (Figure 7D). Author didn't show single experiment to show that OLF form fibrils. ThT is indication for aggregation but no evidence for fibrils.

Material and Methods

At the section of Expression and purification of OLF for NMR experiments.

Average yields from 1 L minimal media were 0.04 mg/L for OLF_{D380A}, 0.02 mg/L for OLF_{I499F} and 2.2 mg/L for WT.

is it 0.02 mg/L correct? NMR required mM quantity.

Can authors give a detailed purification protocol? The protein was expressed as fusing with MBP. Does all experiments perform after or before MBP cleavage?

Response to Reviewers

Reviewer #1

..The authors describe a breadth of detailed results that may help unravel the still obscure complexities of how gain-of-function mutations promote toxic misfolding and aggregation in disease. The differences for D380A and I499F are particularly noteworthy. There are multiple points that need to be addressed to further define and validate the results and their interpretation.

We thank R1 for their positive impression of our study and we appreciate the attention to detail below to strengthen the manuscript.

1. The proposed model for structure disruption is that Ca is lost first; this step needs to be further validated. There is evidence that Ca binding is diminished in D380A, consistent with mutation of this key Ca-liganding residue. Could there also be perturbations to sodium binding that impact the protein behaviour? Are WT and I499F fully metal bound? Additional information is needed on the effect of urea on metal binding, for WT, D380A, and I499F. For example, a denaturation curve monitored by CD and/or NMR, with more urea concentrations than in Figure 4 (and/or for different metal concentrations), might show evidence for multiple transitions, i.e. loss of metal binding followed by denaturation. It is important to include urea denaturation data for the mutants (as for WT in Figure 2D,C).

Thank you for these excellent points regarding missing direct evidence for Ca²⁺ loss as an early step in OLF unfolding. We considered additional NMR experiments, but already at our lowest urea concentration for WT, 1 M, the Ca²⁺ ligand D478 is one of just a few residues altered. CD is not sensitive enough to detect changes in tertiary structure as a function of urea, and fluorescence chelators we tested were quenched as a function of urea so to address this point we conducted ICP-OES on I499F, using the methods and facility as in PMID 23129764 (where WT and D380A were first reported). The results show that the Ca²⁺ site in I499F is occupied at a molar ratio of 0.2:1 whereas WT OLF is occupied to the same extent as we report in prior publications (0.7-0.8:1, PMID 35831671 and 35831671). These results strengthen the conclusion that the Ca site is perturbed in a partially folded OLF that in principle remains competent for Ca²⁺ binding (i.e. I499F). In sum, although there is ample evidence for perturbation of the Ca²⁺ site, in our revision we toned down language that specifically mentions Ca²⁺ release as a trigger for misfolding because we were unable to explicitly nail down the release of Ca²⁺ in the urea-containing samples.

Manuscript Changes:

Abstract p.2: Whether initiated with mutation or chemical perturbation, unfolding propagates outward to the propeller surface propeller.

Results p.7, section “Partially folded moderate (OLF^{D380A}) and severe (OLF^{I499F}) disease variants...: Analysis of Ca²⁺ levels in OLF^{I499F} by inductively coupled plasma optical emission spectroscopy (ICP-OES) reveals ~20% occupancy compared to OLF^{WT} under buffer conditions used in this study and elsewhere^{23,26} (Supp. Table S2). Thus, despite still harboring all residues necessary for Ca²⁺ chelation, the metal center in OLF^{I499F} is not fully occupied and thus predominantly non-native.

Results p. 21, section “Unfolding from the inside out..”: Interestingly, other calcium-coordinating ligands like D380 and N428 experience minimal perturbations in both the 1 and 2 M spectra. The most likely interpretation of this is that Ca²⁺ binding are retained to some extent during urea-induced unfolding, a notion that is supported by MD simulations of OLF^{I499F} and OLF^{D380A} in which metal ions remain at least partially bound (Supp. Fig. S10) as well as by crystal structures of certain metal ligand stable variants tested (Supp. Fig. S11) in which a metal binds, but using an alternative ligand arrangement.

2. The conclusion that amyloid aggregation competes with chemical unfolding but only below the threshold where OLF loses discrete tertiary structure needs clarification and further consideration. Since urea and mutations promote aggregation under conditions where the protein is presumably predominantly folded, how does this competition occur? Also, the prominent random coil signals in the spectrum of I499F (Figure 5) could be due to disruption of the monomer structure, as proposed, or be caused by protein self-association. Are the spectra of the various proteins constant with time or varying due to changing proportions of monomer and self-associated species?

We appreciate this comment. Our urea experiments suggest that when OLF^{WT} is predominantly folded, the partially folded elements trigger the templated aggregation. It is the population of low-lying

excited states that are accessed with urea or mutation, and these are aggregation prone due to accessing the APRs. Self-association cannot readily account for the molten globule-like spectrum for I499F. Even though this variant is prone to aggregation we did not see a reduction in intensity in the spectrum. When purified, I499F (and WT, D380A) elute from size exclusion as monomers. For WT OLF, we have evidence for only very weak dimerization (PMID 31484937), using crosslinking where the OLF concentrations used were similar to the NMR studies here. Thus, we do not think that oligomerization or self-association in a soluble state can account for what is observed here.

Manuscript changes:

Results p. 13 section “Analysis of the moderate disease mutant..” ..eight residues in the previously identified¹⁹ APR stretch P3 (N428, A429, F430, I431, C433, L436, Y437, T438), experience CSP. Interestingly, P1 residues remain largely WT-like, indicating they retain a chemical environment that is similar to OLF^{WT}. This observation may explain why D380A forms P3-like fibrils¹⁹: if P3 residues are mobile and disordered and not sequestered as in WT, they are available to template fibril formation.

Results p. 16 section “Evaluation of the severe disease mutant..” In contrast to OLF^{D380A}, APRs associated with amyloid formation were affected similarly, with three residues disappearing each within P1 (L334, Y335, F336) and P3 (A429, C433, Y437).

Results p. 22 section “Unfolding from the inside out..” In summary, our data suggest a model that explains how OLF misfolding arises, from inside out by unfolding β -propeller (**Figure 7C**), which differentially exposes internal APRs on the pathway to fibrillization (**Supp. Fig. S8**).

Discussion p. 26 full paragraph

3. The effects on protein structure and amyloid aggregation of D380A are proposed to resemble those of 1 M urea on WT, while the I499F effects resemble those of 2 M urea on WT. The authors hypothesize that OLF fibrillization proceeds via access to partially unfolded states. It may be that multiple partially unfolded states are promoted by the toroidal fold of OLF, which might initially unfold not just near the metal binding site but also in the region where the N and C termini associate, in a process that is less cooperative than for typical globular proteins. I recommend adding some discussion of these points to emphasize their significance.

We completely agree unfolding is more complicated than two-state unfolding, and this is emphasized throughout the manuscript. Notably, the N and C terminal molecular clasp is stabilized by the disulfide bond (Cys 245-Cys 433).

Manuscript Changes:

Discussion p. 24 “Even though we do not have complete resonance assignments in the molecular clasp, we can infer that the clasp may be somewhat more resistant to complete unfolding due to a disulfide bond between C433 on blade D and C245 on blade E, but likely experiences perturbations given that D478 within blade E is one of the early residues perturbed with 1M urea.”

4. The minor changes observed by NMR for WT over months might arise from various covalent changes, e.g. deamidation, oxidation, cleavage not apparent by SDS-PAGE. It is important to check for such modifications e.g. by mass spectrometry.

We recruited collaborators to conduct the suggested mass spectrometry analysis on the sample used for NMR (¹⁵N labeled, now 1 year old) and a freshly purified WT OLF (unlabeled) sample for comparison. Chymotrypsin was used for digestion (instead of trypsin) to maximize overall sequence coverage. Oxidation and deamidation were observed at several sites but no clear differences were delineated, most differences between fresh and old sample less were than 10%. The only candidate for a new PTM is deamidation at N480, which was only detected in the old sample, but no conclusions can be drawn because it was measured at low abundance (1 PSM, 2.22% modified). In addition to mass spectrometry we looked again at the 1.3 Å crystal structure, which was solved from a 10 mo old crystal, and no modifications were observed to the domain in the structure.

Manuscript Changes:

Supp. Fig. S13 contains mass spectrometry data analysis. Supplemental data set included with the data.

Materials and Methods (p 36) have been added under section “Sample preparation and LC-MS/MS analysis for posttranslational modifications.”

Results p. 20 section “Dynamics of OLF^{WT} ..” Further, comparing the posttranslational modifications (PTMs), namely, deamidation and methionine oxidation, in the sample used for NMR that had been stored at 4 °C for over a year and freshly purified OLF^{WT} (**Supp. Fig. S13**) reveals just one PTM detected exclusively in the old sample, deamidation at N480, but it was detected at very low abundance (2.22% mean modified %PSM). Eleven PTMs were detected in both samples. The extent of methionine oxidation or deamidation across these common sites in the two samples differ by < 10% in most cases except for N350, N450, and M494 where the difference is less than 17%. In sum, the extremely long half-life for the observed structural change cannot be reconciled by cleavage, degradation, or PTMs and suggests that OLF^{WT} possesses two very similar structural states that are separated by a high energetic barrier.

5. Minor points

- **Please add numbers for the x axis to Figure 1A[sic]**

Figure has been amended.

- **add explanation for the blue spheres in the legend for Figure 6C.**

Figure caption revised to: “Regions that experienced minimal exchange are represented as spheres (right).”

- **I think A and B labels are swapped in Figure 7**

Fixed.

- **please specify whether teal indicates no significant CSP and the quantitative criteria for yellow or teal color.**

In the Figure 7B the yellow color indicates regions of OLF with minimal CSP where resonances were confidently assigned. The teal color indicates assigned WT-OLF resonances that were not able to be confidently transferred across all conditions listed (1 M, 2 M, D380A, and I499F).

Manuscript Changes:

Materials and Methods (p 31): criteria are listed

Figure 7B legend: Updated to be more clear.

- **include explanation of color for bound metals.**

Metals are now labeled in figures

- **Many resonances for WT in Figure 3 may exhibit splitting (e.g. shoulders on T259, S393, I432, D454, maybe nearby minor peaks for G266), or these unlabelled minor signals may arise from residues that are not assigned. Please elaborate on this aspect (e.g. in the SI).**

This is an interesting point, one, unfortunately, that we cannot confidently resolve with our current data. Some of the shoulders are clearly other resonances (see G241 and T448). However for the unassigned resonances we are not confident. Nominally NMR dynamics experiments would illuminate this (ZZ-exchange or similar) but because WT-OLF exhibits unusual long timescale dynamics we are under non-equilibrium conditions limiting this application.

Manuscript Changes:

Materials and Methods (p. 30): We note that some resonances have shoulders which could indicate slow time-scale dynamics. However, the current level of assignments and data quality do not allow for confident assessment of the origins of the less intense resonances that are overlapping on some more intense resonances.

- **Please include additional quantitative details for the principal component analysis of Figure 4 e.g. how many significant components were identified?**

TRENDNMR outputs multiple principal components. However as PC1 represents the most significant principal direction with the largest variance and in this experiment we were deliberately

changing the urea concentration, we anticipate that only PC1 is relevant. Accordingly PC1 shows two-state behavior and generally agrees with the ThT kinetics experiments. PC1 represents 29% contribution of the data. This contribution is listed in the methods (p 31).

- **RMSF values are not shown for the (most mobile) N-terminal residues in Supp. Fig. S6. Please add information on these values (if too large, can state in the legend).**

Residues prior to Gly244 were not modeled due to them being absent from available crystal structures. Note MD simulations have been updated per R3 are now presented in Supp. Fig. S10.

Reviewer #2

..The authors have performed a lot of experiments and while I am no expert in the area in general I agree with the main conclusions but I have some of my own points/comments.

We thank Reviewer 2 for their key critical review of our HDX-MS data and other comments.

6. I have been specifically asked by the editor to look at the HDX-MS aspects of the m/s so I'll start here. The authors are evaluating isotope incorporation directly and not using the more conventional approach comparing the HDX behaviour of one protein by comparison to a reference sample. There is nothing wrong with this approach but I note that the authors have not taken any control measurements to account for back and forward exchange artifacts. Back exchange artefacts in particular can be considerable and lead to false interpretation of experimental data if they haven't been accounted for. As things stand they cannot use the current data and if they want to include HDX-MS dynamics in the article they will have to perform new acquisitions and they have 2 options. First, reacquire the experimental data but with 3 replicates of back exchange controls and 3 replicates for forward exchange controls. Back exchange control samples are fully in-exchanged protein (the same as the experimental protein) acquired as a 15 second labelling experiment. In exchange can be performed by unfolding the protein in excess D2O and then refolding by dialysis against D2O or some equivalent using diafiltration. Alternatively the protein can be incubated in labelling buffer for several days at 37 degrees C this normally exchanges all the 1H for 2H. Ideally aliquots would be taken at several timepoints during incubation to see when the D-incorporation has plateaued but 7-days incubation is normally sufficient. Forward exchange controls constitute a reference acquisition on 1H protein but with the quench made in D2O so that the final D2O concentration in the quench is 50%. Correction of the RFU for each timepoint and each peptide is then made using the following: $RFU_{corr} = (RFU_{obs} - RFU_{fwdexch.}) / (RFU_{backexch.} - RFU_{fwdexch.})$ Second, the authors could use the more conventional difference data approach and in their case obtain experimental data for the WT and one of the variants and report ΔRFU . Note for this they would still have to take back exchange control data and correct the RFU, forward exchange correction is considered unnecessary for this type of data. Note the acquisition of forward and back exchange control data cannot be taken after the fact. The control data must be taken in the same window as the experimental acquisitions.

Thank you so much for this detailed comment, indeed it would have been awful to have these improperly interpreted results in any published manuscript. In our revised manuscript, we restrict our comparison to relative differences across WT vs I499F vs D380A, not absolute differences. The HDX-MS results were edited out of an early version of the original manuscript but the experiments conducted within a short time frame in late 2019, under the same experimental conditions, instruments, etc. We discussed our data with Dr. Renhao Li, now an author, whose lab conducts HDX-MS on proteins on a frequent basis. We believe our interpretation of the HDX-MS data bolsters the initial observation that D380A and I499F adopt structures that are different from WT (which is how the story unfolded for us in real time, before we did any NMR).

We could not redo the experiments to take into consideration absolute back exchange because the core facility we used to do these experiments was taken off line after the pandemic. We ask for R2's understanding regarding this unfortunate complication. Our data have been submitted to ProteomeXchange (Accession PXD045520 Username: reviewer_pxd045520@ebi.ac.uk Password: Y5LjuCZy).

Manuscript changes:

Results p. 7 section “Partially folded moderate (OLF^{D380A}) and severe (OLF^{I499F}) disease..” See new paragraph, the second in this section.

Materials and Methods p. 34 section HDX-MS. See amended paragraph starting with “HDX-MS was performed on OLF^{WT}, OLF^{D380A}, and OLF^{I499F} on three separate days, within a two month period,..”

Data availability p. 38 “The HDX-MS data have been deposited to the ProteomeXchange Consortium with the dataset identifier PXD045520.”

7. Further comments

- **F2 A change colour key remove [urea] line Fixed.**

- **F2 C The test suggests that the differences between 3 and 5 M urea are due to competition between refolding and aggregation and at 5 M aggregation wins. But at higher urea concentrations could the protein just be kinetically trapped in a higher energy unfolded state? It’s probably not, but the authors should indicate refolding time so we can be sure the system is in a steady state. Also find it odd that the protein cannot refold. Once chaotrope is removed the protein is no longer unfolded or folded where is it the authors didn’t state, does it precipitate or form fibres? – clarify.**

Fig. 2C shows the refolding from 3 M and 5 M yields tertiary CD spectra that are both flatlined, indicating that what is left *in solution* is unfolded. The precipitated OLF resulting from dilution from 5 M is not recorded in the instrument. Regarding steady state- the refolding time, e.g. time between diluting out urea to CD measurements, was ~20 minutes for both 3M and 5M. OLF refolded from 3 M urea resembled native, folded OLF in that timeframe (Fig. 2C). For OLF refolded from 5 M urea, more protein crashed out of solution as more time passed, so measurements were taken at early time points where something was left in solution to measure. Regarding the finding that the protein cannot refold, this is not unusual; many proteins require molecular chaperones to fold. Only the simplest model systems are fully reversible.

Manuscript changes:

Results p. 6 section “Amyloid aggregation competes with unfolding”: At room temperature, OLF^{WT} incubated in 3M urea, where tertiary structure is no longer detected by near-UV circular dichroism (CD), can be refolded upon dilution to buffer lacking urea (**Figure 2C**)²⁰. By contrast, when OLF^{WT} incubated in 5M urea is diluted, significant precipitation is visible. Still, some OLF^{WT} remains in solution, at high enough concentration to be measured by near-UV CD, and this species does not have detectable tertiary structure (**Figure 2C**). Thus, refolding OLF^{WT} from urea is possible from the midpoint of unfolding but not at later stages.

Materials and Methods p. 31 section “OLF^{WT} unfolding and refolding by CD: For unfolding, OLF was allowed to equilibrate in 3 M urea for at least 1 h at 4 °C prior to acquisition of CD spectra. For refolding, OLF was equilibrated in either 3 or 5 M urea for at least 15 m at room temperature before being diluted 10-fold with PGF buffer. Spectra were recorded within 20 m of diluting with PGF.

- **F3B can the authors indicate on the figure the residues missing from the xtal structure for quick reference.**

These are now listed the text in the results section for WT assignments (specifically E230-S231, L236-T243 and K503-M504).

- **Line 208 “This result emphasizes the central importance of the Ca²⁺ site in the structural integrity of the OLF propeller.” What do the authors mean by this point can they clarify? Taken as read they mean the site not the ion and that loss of structural integrity at the site perturbs that whole structure. If ion is important in structural integrity the authors should demonstrate this with wild-type but with the addition of a chelator to remove the calcium.**

See comments in response to R1.

- **Figure 7d is confusing. There should be 2 folding landscapes one for monomers and one for different types of multimers with both landscapes being linked by a common monomeric intermediate. The current image doesn’t accurately represent the authors claim that the fibril form is accessed via an intermediate in the current representation the fibril can be access from the fully unfolded state which is**

at odds with the experimental data. The fibril is also lower energy than the native monomer which I don't think is necessary true. 2 landscapes joined by a common intermediate would be much clearer.

Thank you for noticing the error in our figure, it has been updated to better reflect our finding that aggregation from a chemically unfolded state does not lead to aggregation. The relative energies of folded and fibrillized protein have been left as-is as they are in agreement with literature whereby folded proteins are higher energy than fibrils (e.g. see Figure 1 in most review articles, e.g. PMID: 21776078, 23746257).

Reviewer #3

.. This is an interesting study. The most exciting finding is that the unfolding starts in the interior and not exterior of the beta-propeller.

We thank R3 for their overall positive impression of our study and excellent suggestions.

8. Based on my solid knowledge of NMR spectroscopy (though I am not an expert in it), I would say that the NMR experiments (and also H-D exchange mass spectrometry) were performed very carefully and at a high level of standards.

HDX-MS was performed carefully but in the originally submitted manuscript, was inadvertently presented in a misleading way; see comment 6 (R2); this has been fixed by comparative studies between WT and I499F and D380A and its discussion moved to earlier in the MS.

9. However, the MD simulations are below standard in terms of quantity and approach. First, the simulations should be repeated at conditions mimicking the different experimental conditions by adding urea to the WT and introducing mutations D380A and I499A. Moreover, either much longer (10 microseconds) or enhanced simulations (REMD) should be conducted to observe structural changes that could be related to the experimental observations. I highly recommend to use another MD engine than NAMD as NAMD is too slow for systems below 500,000 atoms. Gromacs or Amber would be better MD codes for this purpose.

We repeated the simulations for the WT and added simulations for the D380A and I499F mutants (Supp. Fig. S10). These were run in triplicate for 10 μ s each (90 μ s in total). As recommended by the reviewer, we used Amber for these simulations. For comparison, we note that using the recommended settings for the CHARMM force field (most notably a 12-Å cutoff), Amber produced ~330 ns/day while NAMD3 on GPUs produced ~250 ns/day.

10. Because of the rather short MD simulations, the results on the protein dynamics presented in Fig. 6 are the weakest part of the study, as there is a missing link between the short- and long-time scale dynamics. The aim (for a manuscript in Nat. Commun.) should be to provide a structural picture that shows how the unfolding starts.

Extending the simulations to 10 μ s revealed additional subtle behaviors compared to the previous 1- μ s runs, most notably increased RMSF in particular regions. The addition of simulations of the two mutants also allowed us to make further comparisons to experimental results. Since a connection between simulation results to longer-time behavior in experiment remains speculative, likely due to the complexity of the system, additional suggested experiments like adding urea, seem premature.

Manuscript changes:

Results p. 18 section "Dynamics of OLF^{WT} from ps-month time scale": see new paragraph

11. more specific comments:

- I. 71: Introduce ER (even though it is a common abbreviation)
Fixed

- Fig. 2: i) The color scale is not optimal as the different shades of blue are difficult to distinguish. Please change.

We added a new figure (Supp. Fig. S7) where the spectra are separated out.

- **Fig. 2: ii) Based on Fig. 2B the authors conclude that at high urea concentrations complete unfolding wins and inhibits amyloid aggregation. It is therefore speculated that the latter is initiated from partially folded OLF. However, for Aβ - which is never folded in solution at pH ~7 - it is known that high urea concentrations inhibit amyloid aggregation (see, e.g., Protein Sci. 2004 Nov; 13(11): 2888–2898). If one understands amyloid aggregation as a metafolding problem, it is understandable that this does not take place at high urea concentrations. As long as the structure that finally leads to amyloid aggregation is not fully determined, I find the authors' conclusions with regard to their observations at high urea concentrations not justified.**

Thank you for this point. We address this with changes to the discussion.

Manuscript changes:

Discussion p. 26: Although high urea may stabilize the unfolded state to prevent aggregation, as seen for intrinsically-disordered Aβ⁶⁰, NMR spectra show that above the unfolding midpoint, OLF^{WT} adopts a conformationally heterogeneous structural ensemble consistent with a molten globule-type ensemble (**Figure 4A**), suggesting that in principle, aggregation could still proceed beyond the midpoint of unfolding. Instead, our interpretation of the data point to the possibility that APRs in OLF cannot be fully unfolded for templated fibrillization to occur.

- **Fig. 3 needs to be improved. The residue labels in panel A are not or hardly readable. The size of the figure can be increased to make full use of the page width. The resolution of the figures needs to be increased. And I am sure that also the size of the residue labels could be increased by 1 pt.**

The labels have been made larger. We confirmed that this image was prepared at adequate resolution for the journal. If this manuscript is accepted for publication, we will ensure all figures are up to journal standards.

- **Fig. 4: The meaning of the residues colored in gray in panel C needs to be explained. Please add spheres for Na⁺ and Ca²⁺ to the structures and letters “A” to “E” at the outside of the blades.**

- **Fig. 5: Please add spheres for Na⁺ and Ca²⁺ to the structures, mark the respective mutation site by a star (or alike), and add letters “A” to “E” at the outside of the blades.**

Grey residues (unassigned residues), spheres for the metals, and blade labels have been added to all figures.

- **p. 13: Please add in % how many of the chemical shifts remain almost unchanged upon mutation (and how many cannot be determined).**

Manuscript changes:

Results (p 13) In total, 73 resonance assignments were confidently transferred from OLF^{WT} to the OLF^{D380A} spectrum which represents about 60% of the discreet OLF^{D380A} resonances (see **Methods**).

Results (p 15) Specifically, of the 106 discreet NMR resonances in the OLF^{I499F} spectrum, 51 resonance assignments could be confidently transferred.

- **Fig. 6: The same kind of results should be provided for selected urea concentrations and the mutants. This would better allow to link protein dynamics with unfolding and amyloid aggregation. See also my comment with regard to running further MD simulations.**

We have added HDX-MS results for the mutants for comparison with WT. With this paper as the foundation, our long term goal is to link protein dynamics with unfolding and amyloid aggregation. We are pursuing this with experiments in the lab but this is outside the scope of this already-lengthy manuscript.

Reviewer #4

This paper from the Lieberman lab reports Competition between inside-out unfolding and pathogenic aggregation in an amyloid forming β-propeller. There are several strengths to the work, including unfolding events, that required for amyloid formations. Authors used several techniques including NMR, X-rays, and molecular dynamics simulations to give insight on the OLF domain folding.

We thank R4 for their overall positive impression of the system.

12. Major comment: My major concern regarding the novel of this work. The Crystal Structure of the Myocilin Olfactomedin Domain has been solved (PDB: 4WXQ). It will be more interesting if the authors can determine a novel structure of an intermediated species such as Oligomers or final aggregated form such as amyloid fibril structure of OLF domain.

We agree, but considering the continuum of the trajectory from folded protein (e.g. 4WXQ) to final fibril includes partially folded monomer, we believe the contributions in this manuscript to be a critical piece of the puzzle.

13. I am wondering if OLF domain is fibrilize in human patient. Do the authors have any evidence that OLF present in human patients as fibril species?

To our knowledge, there is just one histochemical study reporting ER-localized accumulation of Y437H mutant myocilin in primary trabecular meshwork cells from a donor eye (PMC6425711). Such donor eyes, with genotyping and complete family medical history, etc are exceedingly rare. As such, for 20 years the field has relied on studies of mutant myocilin from a variety of models.

14. The authors should address the following comments and recommendations on additional critical experiments to make for a stronger study.

- The abstract is long, need to be shortened

The abstract is now 150 words per journal requirements

- Missing reference at the end of this sentence. A recent addition to the list of pathogenic proteins that form amyloid is the myocilin olfactomedin (OLF) domain.

done.

- Missing reference at the end of this sentence. Many different single point mutations, dispersed throughout the sequence, are the strongest genetic link to early onset open angle glaucoma.

done.

- Remove figure 1 from introduction, reference is enough. Figure 1, it is not structure solved in this paper and cannot be in the introduction.

Respectfully, we disagree with the suggestion. In order to introduce the elements of the protein structure germane to the study, we need to have an image of the (previously published) structure in the introduction. We anticipate that this will increase access of our study to non-specialists.

- At the end of introduction, Authors required to bring the message of the question need to be answered in this paper (this is not clear here).

We apologize for the confusion, and have edited the manuscript accordingly.

Manuscript changes:

Introduction, p. 4: To better comprehend the proclivity of OLF toward amyloid aggregation, here we probed the molecular details of its misfolding and aggregation using structural and biophysical tools. We used both low- and high-resolution techniques to probe the mechanism by which OLF^{WT} initially transitions from a folded to a partially unfolded, aggregation prone state, as well as the relationship between partially unfolded states adopted by OLF^{WT} and those adopted by disease-causing mutants.

- Authors reported that they monitored kinetics of amyloid formation in the presence of varying concentrations of urea (Figure 2A, S1B). It will be interested to see under Electron Microscope using negative stain, the aggregation products (Oligomers, fibrils, mature fibrils and amorphous aggregates)

Representative images have been added as Supp. Fig. S1. In other studies e.g. PMC3946817, PMC3323732 we have shown fibril morphologies extensively by TEM and AFM.

- In Figure 1S, it looks like tube with 0 urea, has some aggregation as well. Can author explain? This is just glare from the light in the picture, there was no (yellow tinged) pellet in the tube.

- In Figure 2C, CD spectra of the untreated sample doesn't seem folded well. I am wondering that the protein is not completely pure or folded after purification. Authors need to show CD spectrum starting from 190 nm, possible there are more interesting spectrum to show.

CD spectra in the near-UV (~250 – 320 nm) assesses protein tertiary structure, a fingerprint unique to each protein. Far-UV (~190 – 300 nm) CD spectra, which assess secondary structure, are not included in this work due to the fact that we previously did not see any major differences in secondary structure among variants by far-UV (see PMC3946817).

- The ThT signal, doesn't confirm that the protein can form amyloid fibrils, many ThT signal can also produce from amorphous protein aggregation. Authors should use EM to prove that the OLF can form amyloid fibrils

We have shown morphologies extensively by TEM and AFM in prior publications, e.g. PMC3946817, PMC3323732. We have now added TEM images of fibrils grown in urea in Supp. Fig. S1.

- Authors reported at urea concentrations >2 M, the unfolded state is dominant and limits the ability to confidently map assignments and track CSP. I would suggest that Authors need to examine the conformation state under EM first, and see if these conformations are stable structure, they can dialysis the protein against the suitable buffer and recorded the data. It is also possible that at urea concentrations >2 M, the OLF generate number of heterogenous conformations that limit the assignments.

Current cryoEM methods have difficulty resolving "small" proteins. OLF is a 30 kDa protein, well below the size limit to resolve the conformational ensemble. Our data are most consistent with multiple unfolded states, as stated in the manuscript.

- Authors reported that this observation may explain why D380A forms P3-like fibrils: if P3 residues are mobile and 214 disordered and not sequestered as in WT, they are available to template fibril formation. There is no experimental evidence that show that D380A forms can form fibrils.

We apologize for the confusion. We showed D380A forms circular fibrils via AFM in PMC3946817. The reference was added to the end of this sentence in the results section, "This observation may explain why D380A forms P3-like fibrils²⁰"

- Authors reported that they solved a 1.27 Å crystal structure of OLF (Figure 6A, Supp. Table S3). The new structure shares overall features with our previously reported ~2 Å resolution structures. The resolution is higher and better but still similar to previous structure, 2 Å structure. I feel that there no need to solve previously published structure. It will be more interested to solve the same structure in the presence of ligand or inhibitors.

The purpose of the near-atomic resolution structure is to allow us to get more accurate B-factors, which help provide better insight into picosecond time frame dynamics.

- Authors claim that OLF fibrillization in many parts in the paper, especially in the discussion section (For OLF, 439 the fibrillization and chemical unfolding pathways are in significant competition at 37 °C 440 (Figure 7D). Author didn't show single experiment to show that OLF form fibrils. ThT is indication for aggregation but no evidence for fibrils.

We have shown morphologies extensively by TEM and AFM in prior publications, e.g. PMC3946817, PMC3323732. We have now added TEM images of fibrils grown in urea in Supp. Fig. S1D.

- Average yields from 1 L minimal media were 0.04 mg/L for OLF D380A, 0.02 mg/L for OLF I499F and 2.2 mg/L for WT. is it 0.02 mg/L correct? NMR required mM quantity.

Indeed, the yield for I499F is that low and we struggled to produce this sample. 60 g of cell paste (4 different growths and a week of lysing) was purified for a single NMR sample of low uM concentration. Modern NMR spectrometers and cryogenically cooled probes are compatible with this range of protein concentration. Our data collection was optimized for the lower concentration.

- Can authors give a detailed purification protocol? The protein was expressed as fusing with MBP. Does all experiments perform after or before MBP cleavage?

The methods have all been published previously (e.g. PMID 25524706, 24333014, among others) and yes, MBP is stringently removed for experiments herein. We have added back the information about MBP removal in materials and methods.

Manuscript changes:

Materials and Methods p. 27: section “Expression and purification of OLF for NMR experiments”: see additional description of purification protocol and references cited.

REVIEWERS' COMMENTS

Reviewer #1 (Remarks to the Author):

I am, on the whole, satisfied with the authors' responses to the points I raised. I agree with the authors that the evidence points to different mechanisms of aggregation for D380A and I499F. However, I do still have some reservations about the general claim of progressive 'inside out' unfolding, with increasing urea concentration and from D380A to I499F. If this claim is true, then presumably Ca/Na binding would be weakened with increasing urea and from D380A to I499F? Adding such evidence could much strengthen the paper. I am not sure though that the explanation will necessarily be that simple.

I defer to the expertise of the other reviewers for the points they raised. I appreciate the additional work the authors have conducted to address the points raised by all the reviewers. Collectively, the revisions have significantly strengthened the paper. The further MD still does not provide much mechanistic information, but that may not be surprising as the pertinent dynamics may be slow. Overall, the authors have conducted a very extensive investigation that significantly advances understanding of myocilin aggregation. These studies on a relatively large and complex toroidal protein are important for unravelling how myocilin aggregation may cause glaucoma and for a more general understanding of protein aggregation mechanisms in disease.

Reviewer #3 (Remarks to the Author):

Lieberman and co-workers carefully addressed all reviewer comments.

With respect to the MD simulations, I can state that the ten times longer simulations included in the revised manuscript are in much better agreement with the HDX-MS data (now shown in Fig. S3). However, the authors should have made better use of the opportunity to derive more information from the simulation data to explain the HDX-MS results, i.e., to determine the origin of the increased motions observed for some of the blades. This will require looking at individual amino acids, the possible breakdown of intramolecular interactions, correlated motions, and the like.

I would like to mention that I agree with the authors that Fig. 1 should remain in the Introduction.

Reviewer #4 (Remarks to the Author):

Authors improved the revised version. I have no further comments.

RESPONSE TO REVIEWERS

Reviewer #1 (Remarks to the Author):

I am, on the whole, satisfied with the authors' responses to the points I raised. I agree with the authors that the evidence points to different mechanisms of aggregation for D380A and I499F. **However, I do still have some reservations about the general claim of progressive 'inside out' unfolding, with increasing urea concentration and from D380A to I499F. If this claim is true, then presumably Ca/Na binding would be weakened with increasing urea and from D380A to I499F?** Adding such evidence could much strengthen the paper. I am not sure though that the explanation will necessarily be that simple.

I defer to the expertise of the other reviewers for the points they raised. I appreciate the additional work the authors have conducted to address the points raised by all the reviewers. Collectively, the revisions have significantly strengthened the paper. The further MD still does not provide much mechanistic information, but that may not be surprising as the pertinent dynamics may be slow. Overall, the authors have conducted a very extensive investigation that significantly advances understanding of myocilin aggregation. These studies on a relatively large and complex toroidal protein are important for unravelling how myocilin aggregation may cause glaucoma and for a more general understanding of protein aggregation mechanisms in disease.

We thank Reviewer 1 for recognizing our efforts to improve the manuscript in response to reviewer comments. Regarding lingering reservations regarding inside out unfolding, we agree it's complicated. The most disease-relevant aspect of the manuscript is the misfolding disease-variants and both are completely (D380A) or nearly completely (I499F) depleted of metal. We have also solved numerous OLF variant structures where metal binding is perturbed even as the rest of the structure is largely native. Thus, given the entirety of our data, we think our proposal is reasonable. Future studies will be aimed at further exploring the relationship between chemical unfolding and metal binding in wild-type OLF, as well as conducting metal analysis on additional mutants, to add detail to our understanding of OLF unfolding (independent of disease).

We have added text to the discussion (p. 19, new text highlighted):

“The common feature of inside-out unfolding, in spite of unique partially folded states for OLF^{D380A} and OLF^{I499F}, rationalizes that exposure of internal APRs P1 and/or P3 trigger templated OLF fibrillization. **With this knowledge, we can focus future efforts on the details of the connection between chemical unfolding and metal binding in OLF^{WT}, as well as between misfolding and metal ion binding properties of additional pathogenic mutants, to further our understanding of OLF unfolding.** Finally, these observations raise the possibility that stabilizing neighboring residues to avoid unfolding could prevent aggregation. For example, stabilization with a small molecule or antibody, might be an effective therapeutic approach to abort pathogenic aggregation associated with mutant myocilin.

Reviewer #3 (Remarks to the Author):

Lieberman and co-workers carefully addressed all reviewer comments.

With respect to the MD simulations, I can state that the ten times longer simulations included in the revised manuscript are in much better agreement with the HDX-MS data (now shown in Fig. S3). However, **the authors should have made better use of the opportunity to derive more information from the simulation data to explain the HDX-MS results**, i.e., to determine the origin of the increased motions observed for some of the blades. This will require looking at individual amino acids, the possible breakdown of intramolecular interactions, correlated motions, and the like.

We agree that we did not take full advantage of the updated simulations compared to experiments in our revision. In response, we have now added two new figure panels, one for HDX-MS data (Supp. Fig 3D) and one for MD data (Supp Fig. 10E). Each panel shows pairwise differences between WT and D380A or I499F, mapped onto the structure. We also updated the paragraph describing the MD simulations (p.13 bottom-p.14, pasted below, new text highlighted), with more ways in which the HDX-MS and MD results are in general agreement.

“Second, to detect nano-to-microsecond motions, we carried out atomistic molecular dynamics (MD) simulations on OLF^{WT} (Figure 6b, Supplementary Figure 10). The simulations reveal relatively limited motions over a 10- μ s trajectory, similar to previous results obtained carried out at higher simulation temperature albeit for a shorter time²⁷. The largest root mean squared fluctuations (RMSF) values observed in MD simulations are in loops of blade A (Figure 6b, Supplementary Figure 10). One additional loop in blade D and the long loop connecting blades C and D appear somewhat mobile in the simulation (Figure 6b, Supplementary Figure 10). Some of these motions have been captured previously in OLF^{WT} crystal structures solved from crystals belonging to different lattices¹⁷ as well as stable variants containing a mutation to prevent calcium binding at ligand D478 alone or in combination with mutation of other metal binding ligands (Supplementary Figure 11).¹⁷ Conversely, blades B, C, and D remain largely unchanged throughout the simulation, in agreement with experiments presented here comparing elements that remain intact in OLF^{WT} partially unfolded with urea (Figure 4C, Supplementary Figure 8). Compared to MD simulations of OLF^{WT}, and in general agreement with HDX-MS experiments (Supplementary Figure 3), OLF^{D380A} and OLF^{I499F} exhibit a greater degree of fluctuations as well as unique fluctuations (Supplementary Figure 10). Increased fluctuations are most notable in blades A and E in OLF^{D380A}. Simulations of OLF^{I499F} reveal fluctuations similar to, or even slightly less than, OLF^{WT} (Supplementary Figure 10). Despite this trend, we note that while I499 stably participates in a hydrophobic cluster including L248, V251, I432, and L486 in the OLF^{WT} simulations, F499 breaks out of it. Overall, less fluctuation in the simulation is consistent with less net H/D exchange observed for OLF^{I499F} compared to OLF^{WT}. The main increase in fluctuations in OLF^{I499F} compared to OLF^{WT} is a loop within Blade A, located on the bottom face; this feature is seen also in the OLF^{D380A} simulation, and in HDX-MS for both mutants compared to OLF^{WT} (Supplementary Figure 3). Across simulations, the metal center is often perturbed as well (Supplementary Figure 10). For example, we observe a greater average RMSF in residues comprising P3 either coordinating or near the ions (specifically N428, A429, and F430) in OLF^{D380A} compared to OLF^{WT}, in agreement with CSP observed in this region (Figure 5B). Our inability to observe further structural changes may be due to limited (10- μ s) sampling of the conformational landscape; it may also indicate that mutations alter the folding pathway, which is not explored in simulations starting from the OLF^{WT} folded structure. In summary, on the microsecond time scale simulations suggest fluctuations within OLF are limited largely to loops and the metal center, with distinct perturbations for each mutant.”

I would like to mention that I agree with the authors that Fig. 1 should remain in the Introduction.

Reviewer #4 (Remarks to the Author):

Authors improved the revised version. I have no further comments.